# Dynamically coupling Full Stokes and Shallow Shelf Approximation for marine ice sheet flow using Elmer/Ice (v8.3)

Eef C. H. van Dongen[1,2,3,4], Nina Kirchner[2,5], Martin B. van Gijzen[3], Roderik S. W. van de Wal[4], Thomas Zwinger[6], Gong Cheng[5,7], Per Lötstedt[5,7], and Lina von Sydow[5,7]

[1]Laboratory of Hydraulics, Hydrology and Glaciology, ETHZ, Zurich, Switzerland
[2]Department of Physical Geography, Stockholm University, Stockholm, Sweden
[3]Department of Applied Mathematical Analysis, Delft University of Technology, Delft, The Netherlands
[4]Institute for Marine and Atmospheric Research Utrecht, Utrecht University, The Netherlands
[5]Bolin Centre for Climate Research, Stockholm University, Stockholm, Sweden
[6]CSC-IT Center for Science, Espoo, Finland
[7]Division of Scientific Computing, Department of Information Technology, Uppsala University, Uppsala, Sweden

*Correspondence to:* E. C. H. van Dongen (vandongen@vaw.baug.ethz.ch)

**Abstract.** Ice flow forced by gravity is governed by the Full Stokes (FS) equations, which are computationally expensive to solve due to the non-linearity introduced by the rheology. Therefore, approximations to the FS equations are commonly used, especially when modelling a marine ice sheet (ice sheet, ice shelf and/or ice stream) for $10^3$ years or longer. The Shallow Ice Approximation (SIA) and Shallow Shelf Approximation (SSA) are commonly used but are accurate only for certain parts of an ice sheet. Here, we report a novel way of iteratively coupling FS and SSA that has been implemented in Elmer/Ice and applied to conceptual marine ice sheets. The FS-SSA coupling appears to be very accurate; the relative error in velocity compared to FS is below 0.5% for diagnostic runs and below 5% for prognostic runs. Results for grounding line dynamics obtained with the FS-SSA coupling are similar to those obtained from a FS model in an experiment with a periodical temperature forcing over 3000 years that induces grounding line advance and retreat. The rapid convergence of the FS-SSA coupling shows a large potential in reducing computation time, such that modelling a marine ice sheet for thousands of years should become feasible in the near future. Despite inefficient matrix assembly in the current implementation, computation time is reduced by 32%, when the coupling is applied to a 3D ice shelf.

## 1 Introduction

Dynamical changes in both the Greenland and Antarctic ice sheets are, with medium confidence, projected to contribute 0.03 to 0.20 m of sea level rise by 2081-2100 (Church et al., 2013). The main reason for the uncertainty in these estimates is a limited understanding of ice dynamics. Thus, there is a great need for improvement of ice dynamical models (Ritz et al., 2015). The gravity-driven flow of ice is described by the Full Stokes (FS) equations, amended by a non-linear rheology described by Glen's flow law. Model validation is required over centennial to millennial time scales to capture the long response time of an ice sheet to external forcing (Alley et al., 2005; Phillips et al., 2010; Stokes et al., 2015). However, the computation time and memory required for a FS model to be applied to ice sheets restricts simulations to sub-millenial timescales (Gillet-Chaulet

et al., 2012; Gladstone et al., 2012a; Nowicki et al., 2013; Seddik et al., 2012; Joughin et al., 2014; Seddik et al., 2017). Therefore, approximations of the FS equations are employed for simulations over long timescales, such as the Shallow Ice Approximation (SIA, Hutter, 1983), the Shallow Shelf Approximation (SSA, Morland, 1987; MacAyeal, 1989), Blatter-Pattyn (Pattyn, 2003), and hybrid models (Hindmarsh, 2004; Bernales et al., 2017).

Any ice sheet model accounting for ice shelves needs to resolve grounding line dynamics (GLD). Despite many recent efforts, modelling GLD still poses a challenge in numerical models, as illustrated by the wide range of results obtained in the Marine Ice Sheet Model Intercomparison Project (MISMIP, Pattyn et al., 2012). In MISMIP3d, GLD differ between FS models and SSA models, with discrepancies attributed to so-called higher order terms which are neglected in SSA models but included in FS models (Pattyn et al., 2013). Based on these model intercomparisons, it is advised to use models that include

vertical shearing to compute reliable projections of ice sheet contribution to sea level rise (Pattyn and Durand, 2013). On the other hand, it is not entirely clear how much of the difference in GLD is due to the different numerical treatment of the grounding line problem in shallow models. An updated version of the hybrid SIA/SSA Parallel Ice Sheet Model (PISM) that uses a modified driving stress calculation and subgrid grounding line interpolation showed GLD comparable to a FS model (Feldmann et al., 2014). It should be noted that the experiments in MISMIP3d were idealized, laterally extruded 2D geometries

with quite small sideward disturbances and MISMIP+ (Asay-Davis et al., 2016) may give more insight on realistic situations. Additionally, there is a recent publication that sheds new light to a possible problem with the setup of MISMIP experiments (Gladstone et al., 2018).

     Solving the FS equations over large spatio-temporal domains is still infeasible. However, solvers combining approximations (e.g. SIA or SSA) with the FS equations allow simulation of ice dynamics over long time spans without introducing artifacts

caused by application of approximations in parts of the domain where they are not valid. For instance, Seroussi et al. (2012) coupled FS and SSA, in the framework of the Ice Sheet System Model (ISSM, Larour et al., 2012). They apply the Tiling method which includes a blending zone of FS and SSA. Their result looks promising with respect to both accuracy and efficiency, but is limited to diagnostic experiments. The Ice Sheet Coupled Approximation Levels (ISCAL) method (Ahlkrona et al., 2016) couples SIA and FS by a non-overlapping domain decomposition that dynamically changes with time. ISCAL

is implemented in Elmer/Ice (Gagliardini et al., 2013), an open source finite element software for ice sheet modelling. Here, we present a novel coupling between FS and SSA, also by implementation of a non-overlapping domain decomposition in Elmer/Ice. The domain decomposition changes dynamically with grounding line advance and retreat. GLD are modelled with FS and coupled to SSA on the ice shelf via boundary conditions. The equations discretized by the finite element method are solved iteratively, alternating between the FS and the SSA domain, until convergence is reached.

The extent of present-day ice shelves is limited to approximately 10 % of the area of Antarctica (Rignot et al., 2013). Therefore, one may question the reduction of computational work by applying SSA to model ice shelves in continental scale simulations of marine ice sheets. However, the coupling is targeted to conducting paleo-simulations, for which much larger ice shelves have been present (Jakobsson et al., 2016; Nilsson et al., 2017). In that case, a large part of the interior of a marine ice sheet is modelled with SIA, SSA is applied to the ice shelves and the FS domain is restricted to ice streams and areas around the grounding line.

An overview of the FS and SSA equations governing ice sheet and shelf dynamics in three dimensions (3D) is presented in Sect. 2, together with the boundary conditions. Memory and performance estimates of a FS-SSA coupling, independent of the specific coupling implemented, are provided in Sect. 2.3. Section 3 describes the coupled FS-SSA model, hereafter 'coupled model'. The coupling is applied to a conceptual ice shelf ramp and marine ice sheet in Sect. 4. The simulation of a 3000 years long cycle of grounding line advance and retreat (described in Sect. 4.2.2) shows the robustness of the coupling.

## 2 Governing equations of ice flow

Ice is considered as an incompressible fluid, such that mass conservation implies that the velocity is divergence-free,

$$\nabla \cdot \boldsymbol{u} = 0, \tag{1}$$

where $\boldsymbol{u} = (u, v, w)^T$ describes the velocity field of the ice with respect to a Cartesian coordinate system $(x, y, z)^T$ where $z$ is the vertical direction. For ice flow, the acceleration term can be neglected in the Navier-Stokes equations (Hutter, 1982). Therefore, the conservation of linear momentum under the action of gravity $\boldsymbol{g}$ can be described by

$$-\nabla p + \nabla \cdot \left( \eta \left( \nabla \boldsymbol{u} + (\nabla \boldsymbol{u})^T \right) \right) + \rho \boldsymbol{g} = \boldsymbol{0}, \tag{2}$$

where $\nabla$ is the gradient operator, $p$ pressure, $\eta$ viscosity, $\rho$ ice density and $\boldsymbol{g}$ denotes gravity. Letting $\boldsymbol{\sigma}$ denote the stress tensor, pressure $p$ is the mean normal stress ($p = -1/3 \Sigma_i \sigma_{ii}$), and $\mathbf{D}(\boldsymbol{u})$ is the strain rate tensor, related by

$$\boldsymbol{\sigma} = 2\eta \mathbf{D}(\boldsymbol{u}) - p\mathbf{I} = \eta \left( \nabla \boldsymbol{u} + (\nabla \boldsymbol{u})^T \right) - p\mathbf{I}, \tag{3}$$

where $\mathbf{I}$ is the identity tensor. Together, Eq. (1) and Eq. (2) are called the Full Stokes (FS) equations. Observations by Glen (1952) suggest that the viscosity depends on temperature $T$ and the effective strain rate $D(\boldsymbol{u})$,

$$\eta(\boldsymbol{u}, T) = \frac{1}{2} \mathcal{A}(T)^{-\frac{1}{n}} D(\boldsymbol{u})^{\frac{1-n}{n}}, \tag{4}$$

$$D(\boldsymbol{u}) = \sqrt{\frac{1}{2} \left( \left( \frac{\partial u}{\partial x} \right)^2 + \left( \frac{\partial v}{\partial y} \right)^2 + \left( \frac{\partial w}{\partial z} \right)^2 \right) + \frac{1}{4} \left( \left( \frac{\partial u}{\partial y} + \frac{\partial v}{\partial x} \right)^2 + \left( \frac{\partial u}{\partial z} + \frac{\partial w}{\partial x} \right)^2 + \left( \frac{\partial v}{\partial z} + \frac{\partial w}{\partial y} \right)^2 \right)}, \tag{5}$$

where Glen's exponent $n = 3$. The fluidity parameter $\mathcal{A}$ increases exponentially with temperature as described by the Arrhenius relation (Paterson, 1994). This represents a thermodynamically coupled system of equations. However, in the current study, we focus on the mechanical effects and a uniform temperature is assumed. Due to the velocity dependence of the viscosity in Eq. (4), the FS equations form a non-linear system with four coupled unknowns, which is time consuming to solve. Therefore, many approximations to the FS equations have been derived in order to model ice sheet dynamics on long timescales, see Sect. 2.1.

### 2.1 Shallow Shelf Approximation

Floating ice does not experience basal drag, hence all resistance comes from longitudinal stresses or lateral drag at the margins. For ice shelves, the Shallow Shelf Approximation (SSA), has been derived by dimensional analysis based on a small aspect

ratio and surface slope (Morland, 1987; MacAyeal, 1989). This dimensional analysis shows that vertical variation of $u$ and $v$ is negligible, such that $w$ and $p$ can be eliminated by integrating the remaining stresses over the vertical and applying the boundary conditions at the glacier surface and base (described in Sect. 2.2). Then, the conservation of linear momentum, Eq. (2), simplifies to

$$\nabla_h \cdot (2\bar{\eta}(\mathbf{D_h}(\boldsymbol{u}) + \text{tr}(\mathbf{D_h}(\boldsymbol{u}))\mathbf{I})) = \rho g H \nabla_h z_s \tag{6}$$

where the subscript $h$ represents the components in the $x-y$ plane, $\bar{\eta}$ the vertically integrated viscosity, $H$ the thickness of the ice shelf and $z_s$ the upper ice surface, see Fig. 1. The effective strain rate in Eq. (5) simplifies to

$$D_h(\boldsymbol{u}) = \sqrt{\left(\frac{\partial u}{\partial x}\right)^2 + \left(\frac{\partial v}{\partial y}\right)^2 + \frac{\partial u}{\partial x}\frac{\partial v}{\partial y} + \frac{1}{4}\left(\frac{\partial u}{\partial y} + \frac{\partial v}{\partial x}\right)^2}, \tag{7}$$

where $w$ is eliminated using incompressibility, Eq. (1). The SSA equations are still non-linear through $\bar{\eta}$, but since $w$ and $p$ are eliminated and vertical variation of $u$ and $v$ is neglected, the 3D problem with 4 unknowns is reduced to a 2D problem with 2 unknowns. Therefore, the SSA model is less computationally demanding than FS. The horizontal velocities are often of main interest, for example when results are validated by comparison to observed horizontal surface velocity. If desirable, the vertical velocity can be computed from the incompressibility condition.

## 2.2 Boundary conditions and time evolution

The coupling is applied to a marine ice sheet, with bedrock lying (partly) below sea level (see Fig. 1), and involves boundaries in contact with the bedrock, ocean and atmosphere. The only time dependency is in the evolution of the free surfaces.

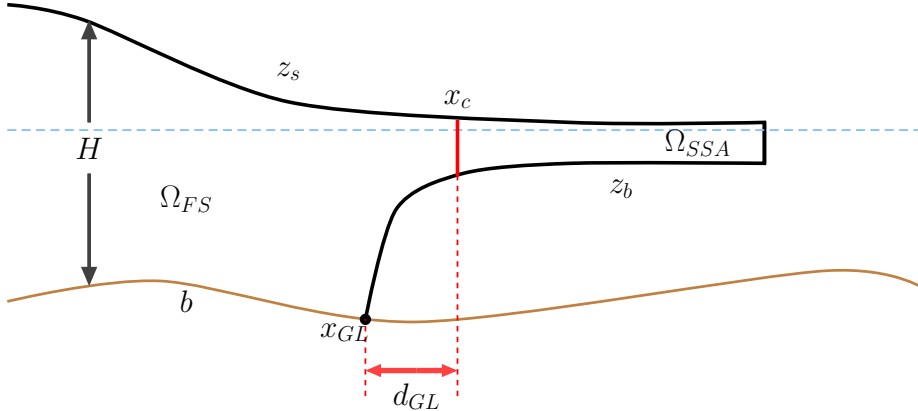

**Figure 1.** Overview of the notations and domain decomposition for a conceptual marine ice sheet. The vertical scale is exaggerated. The sea level at $z = 0$ is dashed blue and the interface between the FS and SSA domains is solid red. The bed elevation is denoted by $b$, the coupling interface by $x_c$ and the grounding line by $x_{GL}$. The distance between $x_c$ and $x_{GL}$, defined in Eq. (17), is denoted $d_{GL}$.

### 2.2.1 Bedrock

Where the ice is grounded (in contact with the bedrock), the interaction of ice with the bedrock is commonly represented by a sliding law $f(\boldsymbol{u}, N)$, that relates the basal velocity $\boldsymbol{u}_b$ and effective pressure $N$ to the basal shear stress as

$$(\boldsymbol{t}_i \cdot \boldsymbol{\sigma} \cdot \boldsymbol{n})_b \;=\; f(\boldsymbol{u}, N)\boldsymbol{u} \cdot \boldsymbol{t}_i, \;\; i = 1, 2, \tag{8}$$

$$(\boldsymbol{u} \cdot \boldsymbol{n})_b + a_b \;=\; 0, \tag{9}$$

where $\boldsymbol{t}_i$ are the vectors spanning the tangential plane, $\boldsymbol{n}$ is the normal to the bed, and $a_b$ describes basal refreezing or melt. A sliding law suggested by Budd et al. (1979) is assumed, which depends on $\boldsymbol{u}_b$ and the height above buoyancy $z_*$ such that

$$f(\boldsymbol{u}, N) \;=\; -\beta |\boldsymbol{u}_b|^{\frac{1}{n}-1} z_*(N). \tag{10}$$

Here, the sliding parameter $\beta$ is constant in time and space. In line with Gladstone et al. (2017), instead of modeling $N$, a hydrostatic balance is assumed to approximate $z_*$, implying a sub-glacial hydrology system entirely in contact with the ocean,

$$z_*(H) = \begin{cases} H & \text{if } z_b \geq 0, \\ H + z_b \frac{\rho_w}{\rho} & \text{if } z_b < 0, \end{cases} \tag{11}$$

where $z_b$ is the lower ice surface, $\rho_w$ the water density and the sea level is at $z = 0$. Equation (11) implies that $z_*$ equals zero when the flotation criterion (Archimedes' principle) is satisfied, i.e. where

$$z_s = \left(1 - \frac{\rho}{\rho_w}\right)H, \;\; z_b = -\frac{\rho}{\rho_w}H. \tag{12}$$

### 2.2.2 Ice-ocean interface

As soon as the seawater pressure $p_w$ at the ice base $z_b$ is larger than the normal stress exerted by the ice at the bed, the ice is assumed to float. For a detailed description of the implementation of the contact problem at the grounding line in Elmer/Ice, see Durand et al. (2009). At the ice-ocean interface, the tangential friction is neglected ($f(\boldsymbol{u}, N) \equiv 0$ in Eq. (8)) and

$$\boldsymbol{\sigma} \cdot \boldsymbol{n} = -p_w \boldsymbol{n} \text{ where } p_w(z) = -\rho_w g z \text{ if } z \leq 0, \tag{13}$$

and $\boldsymbol{\sigma} \cdot \boldsymbol{n} = 0$ above sea level ($z > 0$). Calving at the seaward front of the ice shelf is not explicitly modelled but the length of the modelling domain is fixed and ice flow from the shelf out of the domain is interpreted as a calving rate.

### 2.2.3 Surface evolution

Ice surface (assumed stress-free, $\boldsymbol{\sigma} \cdot \boldsymbol{n} = 0$) and ice base at $z_s$ and $z_b$ behave as free surfaces according to

$$\frac{\partial z_{s/b}}{\partial t} + u_{s/b}\frac{\partial z_{s/b}}{\partial x} + v_{s/b}\frac{\partial z_{s/b}}{\partial y} = w_{s/b} + a_{s/b}, \tag{14}$$

where $a_{s/b}$ is the accumulation ($a_{s/b} > 0$) or ablation ($a_{s/b} < 0$) in meter ice equivalent per year, at the surface or base, respectively. By vertical integration of the incompressibility condition, Eq. (1), $w$ can be eliminated using Leibniz integration

rule and substituting the free surface equations, Eq. (14), which yields the thickness advection equation

$$\frac{\partial H}{\partial t} + \frac{\partial H\bar{u}}{\partial x} + \frac{\partial H\bar{v}}{\partial y} = a_s - a_b, \tag{15}$$

where $\bar{u}, \bar{v}$ are the vertically integrated horizontal velocities.

## 2.3 Memory and performance estimates of a FS-SSA coupling

The reduction of the memory required for a FS-SSA coupling by domain decomposition, compared to a FS model, can be estimated. This estimate is independent of the specific implementation of the coupling between the domains and concerns only the most ideal implementation in which no redundant information is stored. The main advantage of the SSA model is that $\boldsymbol{u}_{SSA}$ is independent of $z$, such that the SSA equations can be solved on a part of the domain with a mesh of one dimension fewer. Besides that, there are fewer unknowns since $p$ and $w$ are eliminated. An additional advantage of eliminating $p$ is that the resulting system is mathematically easier to discretise and solve. In particular, difficulties related to a stable choice for the basis functions for the pressures and velocities are avoided (see e.g., Helanow and Ahlkrona, 2018)) and there is no need for specialised iterative solution techniques to solve the so-called saddle-point problem that the FS equations pose (see Benzi et al., 2005).

Suppose that the computational domain $\Omega$ is discretized with $N_z$ nodes regularly placed in the $z$ direction, $N_h$ nodes in a horizontal footprint mesh and decomposed in two parts ($\Omega_{SSA}$ and $\Omega_{FS}$, see Fig. 1). The fraction of nodes in $\Omega_{SSA}$ is denoted $\theta$ with $0 < \theta < 1$. The number of nodes in $\Omega_{FS}$ is then approximately $(1-\theta)N_h N_z$ and in $\Omega_{SSA}$ it is $\theta N_h$, neglecting shared nodes on the boundary. For a 3D physical domain, SSA has 2 unknowns ($u$ and $v$) and FS has 4 unknowns ($u, v, w$, and $p$). Hence, the memory needed to store the solution with a coupled model is proportional to $2N_h(\theta + 2(1-\theta)N_z)$. For a 2D simulation in the $x-z$ plane, where FS has 3 unknowns and SSA only 1, the memory is proportional to $N_h(\theta + 3(1-\theta)N_z)$. The memory requirement for a physical domain in $d$ dimensions, reduces to

$$q_{\text{var}} = \frac{\text{coupled model memory}}{\text{FS model memory}} = 1 - \theta + \frac{\theta}{(5-d)N_z}, \ d = 2, 3, \tag{16}$$

when part of the domain is modelled by the SSA equations. The memory requirements for mesh related quantities reduces to $q_{\text{mesh}} = 1 - \theta + \theta/N_z$ in both 2D and 3D. The quotients $q_{\text{var}}$ and $q_{\text{mesh}}$ are close to $1 - \theta$ if $N_z \gtrsim 10$.

The computational work is more difficult to estimate *a priori* since it depends on the implementation of the coupling. The dominant costs are for the assembly of the finite element matrices, the solution of the nonlinear equations, and an overhead for administration in the solver. The work to assemble the matrices grows linearly with the number of unknown variables. Suppose that this work for FS in 3D is $4C_{FS}N_h N_z$ in the whole domain, for FS $4C_{FS}(1-\theta)N_h N_z$ in $\Omega_{FS}$, and for SSA $2C_{SSA}\theta N_h$ in $\Omega_{SSA}$. The coefficients $C_{FS}$ and $C_{SSA}$ depend on the basis functions for FS and SSA and the complexity of the equations. The reduction in assembly time for the matrix is $q_{ass} = 1 - \theta + C_{SSA}\theta/2C_{FS}N_z$. If $C_{FS} \approx C_{SSA}$ then the reduction is approximately as in Eq. (16). The same conclusion holds in 2D. Therefore, the reduction of that part is estimated to be similar to the reduction in Eq. (16).

## 3 Method for coupling FS and SSA

All equations are solved in Elmer/Ice (Gagliardini et al., 2013) using the Finite Element Method (FEM). First the velocity $\boldsymbol{u}$ (using FS or SSA) is solved for a fixed geometry at time $t$. The mesh always has the same dimension as the physical modeling domain, but $\boldsymbol{u}_{SSA}$ is only solved on the basal mesh layer, after which the solution is reprojected over the vertical axis. Then, the geometry is adjusted by solving the free surface and thickness advection equations using backward Euler time integration. The non-linear FS and SSA equations are solved using a Picard iteration. The discretized FS equations are stabilized by the residual free bubbles method (Baiocchi et al., 1993), the recommended stabilization method in Gagliardini and Zwinger (2008). First, the coupling for a given geometry is presented, followed by the coupled surface evolution, both summarized in Algorithm 1.

The FS domain $\Omega_{FS}$ contains the grounded ice and a part of the shelf around the grounding line, see Fig. 1. The SSA domain $\Omega_{SSA}$ is restricted to a part of the ice shelf and starts at the coupling interface $\boldsymbol{x}_c$ at the first basal mesh nodes located at least a distance $d_{GL}$ from the grounding line $\boldsymbol{x}_{GL}$, such that

$$||\boldsymbol{x} - \boldsymbol{x}_{GL}|| := \sqrt{(x - x_{GL})^2 + (y - y_{GL})^2 + (z - z_{GL})^2} \geq d_{GL} \text{ for all } \boldsymbol{x} \text{ in } \Omega_{SSA}. \tag{17}$$

### 3.1 Boundary conditions at the coupling interface

Horizontal gradients of the velocity are not neglected in the SSA equations (unlike in the SIA, Hutter, 1983). Thus, not only FS and SSA velocities have to match but also their gradients, in order to allow a coupling of the two. Therefore, one cannot solve one system of equations independently, for use as an input to the other system, as done for a one-way coupling (e.g., Ahlkrona et al., 2016). Instead, the coupling of FS and SSA is solved iteratively, updating the interaction between FS and SSA velocities in each iteration to obtain mutually consistent results. SSA governed ice shelf flow is greatly influenced by the inflow velocity from the FS domain. Therefore, we start the first iteration of the coupled model by solving the FS equations. A boundary condition is necessary at $\boldsymbol{x}_c$, we assume that the cryostatic pressure acts on $\Omega_{FS}$ at $\boldsymbol{x}_c$,

$$\boldsymbol{\sigma}_{FS} \cdot \boldsymbol{n}(\boldsymbol{x}_c, z) = \rho g(z_s - z)\boldsymbol{n}, \tag{18}$$

where $\boldsymbol{n}$ is normal to the coupling interface $\boldsymbol{x}_c$. The FS velocity at $x_c$ provides a Dirichlet inflow boundary condition to the SSA equations. Then, the Neumann boundary condition in Eq. (18) has to be adjusted based on the ice flow as calculated for $\Omega_{SSA}$. This is done using the contact force denoted by $\boldsymbol{f}_{SSA}$, as explained below.

The SSA equations are linearized, and by means of FEM discretized. This leads to a matrix representation $\mathbf{A}\boldsymbol{u} = \boldsymbol{b}$, where $\boldsymbol{u}$ is the vector of unknown variables (here horizontal SSA velocities). In FEM terminology, the vector $\boldsymbol{b}$ that describes the forces driving or resisting ice flow is usually called the body force and $\mathbf{A}$ the system matrix (Gagliardini et al., 2013). In Elmer/Ice, Dirichlet conditions for a node $i$ are prescribed by setting the $i$th row of $\mathbf{A}$ to zero, except for the diagonal entry which is set to be unity, and $\boldsymbol{b}_i$ is set to have the desired value (Råback et al., 2016). For an exact solution of $\mathbf{A}\boldsymbol{u} = \boldsymbol{b}$, the residual $\boldsymbol{f} = \mathbf{A}\boldsymbol{u} - \boldsymbol{b}$ is zero. If we instead use the system matrix $\mathbf{A}_{SSA}$ obtained without the Dirichlet conditions being set, the resulting residual is equal to the contact force that would have been necessary to produce the velocity described by the Dirichlet boundary condition. Since the SSA equations are vertically integrated, $\boldsymbol{f}_{SSA} = \mathbf{A}_{SSA}\boldsymbol{u}_{SSA} - \boldsymbol{b}_{SSA}$ is the vertically

integrated contact force and needs to be scaled by the ice thickness $H$. In Elmer/Ice, $\boldsymbol{f}_{SSA}$ is mesh dependent and needs to be scaled by the horizontal mesh resolution $\omega$ as well. For 2D configurations, $\omega = 1$. Using $\boldsymbol{f}_{SSA}$ instead of explicitly calculating the stress is advantageous since it is extremely cheap to find the contact force if $\mathbf{A}_{SSA}$ is stored.

To summarize the boundary conditions at $\boldsymbol{x}_c$, for FS, an external pressure is applied,

$$\boldsymbol{\sigma}_{FS} \cdot \boldsymbol{n}(\boldsymbol{x}_c, z) = \rho g(z - z_s)\boldsymbol{n} + \frac{\boldsymbol{f}_{SSA}(\boldsymbol{x}_c)}{\omega H}, \tag{19}$$

where $\boldsymbol{f}_{SSA} := 0$ in the first iteration (for its derivation, see Appendix A). For SSA, a Dirichlet inflow boundary condition

$$\boldsymbol{u}_{SSA}(\boldsymbol{x}_c) = \boldsymbol{u}_{FS}(\boldsymbol{x}_c, z_b), \tag{20}$$

provides the coupling to the FS solution. Here we take the $\boldsymbol{u}_{FS}$ at $z_b$, but any $z$ can be chosen since $\boldsymbol{x}_c$ should be located such
that $\boldsymbol{u}_{FS}(\boldsymbol{x}_c, z)$ hardly varies with $z$. Every iteration, $\boldsymbol{f}_{SSA}$ and $\boldsymbol{u}_{FS}(\boldsymbol{x}_c, z_b)$ are updated until convergence up to a tolerance $\varepsilon_c$.

### 3.2  Surface evolution

The surface evolution is calculated differently in the two domains $\Omega_{FS}$ and $\Omega_{SSA}$. Equation (14) is applied to $\Omega_{FS}$ for the evolution of $z_s$ and $z_b$, avoiding assuming hydrostatic equilibrium beyond the grounding line, since the flotation criterion is not
necessarily fulfilled close to the grounding line (Durand et al., 2009). The thickness advection equation, Eq. (15), is used for $\Omega_{SSA}$, which is advantageous since the ice flux $\boldsymbol{q} = H\boldsymbol{u}_{SSA}$ is directly available (because $\boldsymbol{u}_{SSA}$ does not vary with $z$) and no vertical velocity is needed. Moreover, only one time dependent equation is solved instead of one for the lower and one for the upper free surface. The evolution of the surfaces $z_s$ and $z_b$ for $\Omega_{SSA}$ is then calculated from the flotation criterion, Eq. (12). At $\boldsymbol{x}_c$, $H_{SSA} = H_{FS}$ is applied as a boundary condition to the thickness equation. First the surface evolution is solved for $\Omega_{FS}$,
then $\Omega_{SSA}$ follows.

### 3.3  The algorithm

The iterative coupling for one time step is given by Algorithm 1. First, the shortest distance $d$ to the grounding line is computed for all nodes in the horizontal footprint mesh at the ice shelf base. Then, a mask is defined that describes whether a node is in $\Omega_{FS}, \Omega_{SSA}$ or at the coupling interface $\boldsymbol{x}_c$, based on the user defined $d_{GL}$. Technically, the domain decomposition is based
on the use of passive elements implemented in the overarching Elmer code (Råback et al., 2016), which allow for deactivating and reactivating of elements. An element in $\Omega_{FS}$ is passive for the SSA solver, which means that is not included in the global matrix assembly of $\mathbf{A}_{SSA}$, and vice-versa.

Two kinds of iterations are involved, since computing either $\boldsymbol{u}_{FS,k}$ or $\boldsymbol{u}_{SSA,k}$ for the $k$th coupled iteration also requires Picard iteration by the non-linearity in the viscosity. As the experiments will show, calculating $\boldsymbol{u}_{FS,k}$ dominates the computation
time in the coupled model. The coupled model is therefore more efficient if the total number of FS Picard iterations (the sum of FS Picard iterations over all coupled iterations) decreases. This is accomplished by limiting the number of FS Picard iterations before continuing to compute $\boldsymbol{u}_{SSA,k}$, instead of continuing until the convergence tolerance $\varepsilon_P$ is reached, since it is inefficient

to solve very accurately for $\boldsymbol{u}_{FS,k}$ if the boundary condition at $\boldsymbol{x}_c$ is not yet accurate. Despite interrupting the Picard iteration, the final solution includes a converged FS solution since the coupled tolerance $\varepsilon_c$ is reached. Picard iteration for $\boldsymbol{u}_{SSA,k}$ is always continued until convergence since the computation time is negligible compared to FS.

5     An element may switch from $\Omega_{SSA}$ to $\Omega_{FS}$, for example during grounding line advance. Then, the coupled iteration either starts with the initial condition for $\boldsymbol{u}_{FS}$ if the element is in $\Omega_{FS}$ for the first time, or the latest $\boldsymbol{u}_{FS}(t)$ computed in this element, before switching to SSA.

---

**Algorithm 1** Iteratively coupling FS and SSA for one time step, including surface update.

---

**Initialize:** $k := 0$, $\Omega := (\Omega_{FS}, \Omega_{SSA})$ by restricting $\Omega_{SSA}$ to the ice shelf and requiring $||\boldsymbol{x} - \boldsymbol{x}_{GL}||_h \geq d_{GL}$ for all $\boldsymbol{x}$ in $\Omega_{SSA}$.

**if** $t > 0$ **then**
    Take $\boldsymbol{u}_{FS,0}, \boldsymbol{u}_{SSA,0}, \boldsymbol{f}_{SSA,0}$ from previous time step.

**else**
    $\boldsymbol{u}_{FS,0}, \boldsymbol{u}_{SSA,0}, \boldsymbol{f}_{SSA,0} = 0$.

**end if**

$converged$=false

**while** not $converged$ **do**
    Compute $\boldsymbol{u}_{FS,k+1}$ on $\Omega_{FS}$ with boundary condition $\boldsymbol{\sigma}_{FS,k+1} \cdot \boldsymbol{n}(\boldsymbol{x}_c, z) = \rho g(z - z_s)\boldsymbol{n} + \frac{\boldsymbol{f}_{SSA,k}(\boldsymbol{x}_c)}{\omega H}$ at $\boldsymbol{x}_c$.
    Compute $\boldsymbol{u}_{SSA,k+1}$ on $\Omega_{SSA}$ with boundary condition $\boldsymbol{u}_{SSA,k+1}(\boldsymbol{x}_c) = \boldsymbol{u}_{FS,k+1}(\boldsymbol{x}_c, z_b)$.
    Let $\boldsymbol{f}_{SSA,k+1} = A_{SSA,k+1}\boldsymbol{u}_{SSA,k+1} - \boldsymbol{b}_{SSA,k+1}$.
    $converged = ||\boldsymbol{u}_{FS,k+1} - \boldsymbol{u}_{FS,k}||/||\boldsymbol{u}_{FS,k}|| \leq \varepsilon_c$ **and** $||\boldsymbol{u}_{SSA,k+1} - \boldsymbol{u}_{SSA,k}||/||\boldsymbol{u}_{SSA,k}|| \leq \varepsilon_c$
    $k := k + 1$.

**end while**

Surface evolution by free surface equations (Eq. (14) for $\Omega_{FS}$.

Surface evolution by thickness equation (Eq. (15)) for $\Omega_{SSA}$, with $H_{SSA}(\boldsymbol{x}_c) = H_{FS}(\boldsymbol{x}_c)$.

---

## 4   Numerical experiments

To validate the coupled model, we first verify for a conceptual ice shelf ramp that solutions obtained with the coupled model
10   resemble the FS velocity in 2D and 3D. Then the coupled model is applied to a 2D conceptual marine ice sheet (MIS). Whenever 'accuracy of the coupled model' is mentioned, this refers to the accuracy of the coupled model compared to the FS model. Investigating the accuracy of the FS model itself is outside the scope of this study. No convergence study of the FS model with respect to discretization in either time or space is performed. Instead, equivalent settings are used for the FS and coupled model, such that they can be compared and the FS model is regarded as a reference solution.

### 4.1 Ice shelf ramp

#### 4.1.1 Two dimensional ice shelf ramp

A simplified test case is chosen for which the analytical solution to the SSA equations exists in 2D as described in Greve
and Blatter (2009). It consists of a 200 km long ice shelf (see Fig. 2), with a horizontal inflow velocity $u(0, z) = 100$ m yr$^{-1}$
and a calving front at $x = 200$ km where the hydrostatic pressure as exerted by the sea water is applied. The shelf thickness
linearly decreases from 400 m at $x = 0$ to 200 m at $x = 200$ km, $z_b$ and $z_s$ follow from the flotation criterion, Eq. (12). By
construction, the SSA model is expected to be a good approximation of the FS model. The domain is discretized by a structured
mesh, equidistant nodes on the horizontal axis, and extruded along the vertical to quadrilaterals. All constants used and mesh
characteristics are specified in Table A1.

Three models are applied to this setup, FS-only, SSA-only, and the coupled model, for which the horizontal velocities
are denoted $u_{FS}, u_{SSA}$, and $u_c$ respectively. The relative node-wise velocity differences between $u_{SSA}$ and $u_{FS}$ stay below
0.02% in the entire domain. However, computing time for the SSA solution only takes 3% of that of the FS solution, which is
promising for the potential speedup of the coupled model.

The coupling location is fixed at $x_c = 100$ km, as no grounding line is present to relate $x_c$ to. In the first coupled iteration,
$u_c(x_c, z_b) = 100$ m yr$^{-1}$, while in the final solution $u_{FS}(x_c, z_b) = 4805$ m yr$^{-1}$. The cryostatic pressure applied to $\Omega_{FS}$ at
$x_c$ buttresses the ice flow completely and the force imbalance of the hydrostatic pressure at the calving front does not yet
influence the velocity $u_c$ in $\Omega_{FS}$. In the second iteration, when $\boldsymbol{f}_{SSA}$ is applied, a maximum difference of only 0.3% between
$u_{FS}$ and $u_c$ in the entire domain remains. The coupling converges after three iterations, the velocity $u_c$ and relative difference
compared to FS are shown in Fig. 2. Convergence of the coupled model requires 31 FS Picard iterations compared to 35 for FS-
only. However, assembly time per FS iteration almost doubles in the coupled model compared to the FS model, and assembly
time dominates the computational work in this simplified 2D case. Therefore, the coupled model needs almost twice as much
computation time as the FS model. This issue is due to usage of passive elements and is addressed in the Discussion (Sect. 5).

#### 4.1.2 Three dimensional ice shelf ramp

The 2D ice shelf ramp is extruded along the $y$-axis (see Fig. 3). On both lateral boundaries at $y = 0$ and 20 km, $\boldsymbol{u} \cdot \boldsymbol{n}$=0. All
other boundary conditions remain identical to the 2D case and the coupling interface is located halfway $\boldsymbol{x}_c = (100, y)$ km. First
the solutions of the FS and SSA model in Elmer/Ice will be compared before applying the coupled model.

The limited width of the domain (20 km) in combination with the boundary condition $\boldsymbol{u} \cdot \boldsymbol{n} = 0$ at both lateral sides yields a
negligible flow in the $y$ direction ($v_{FS} < 10^{-8}$ m yr$^{-1}$). Despite differences in the models, the relative difference in $u$ is below
1.5%. Running the experiment with the SSA model takes only 0.8% of the time needed to run it with the FS model.

The maximum relative difference between $\boldsymbol{u}_{FS}$ and $\boldsymbol{u}_c$ is 1.4%, which is of the same order of magnitude as the velocity
difference between FS and SSA. The mean assembly time per FS iteration is 6% higher than in the FS-only model, but the
solution time decreases by 55%. Convergence of the coupled model requires 30 FS iterations compared to 27 for FS-only. The
total computation time decreases by 32%.

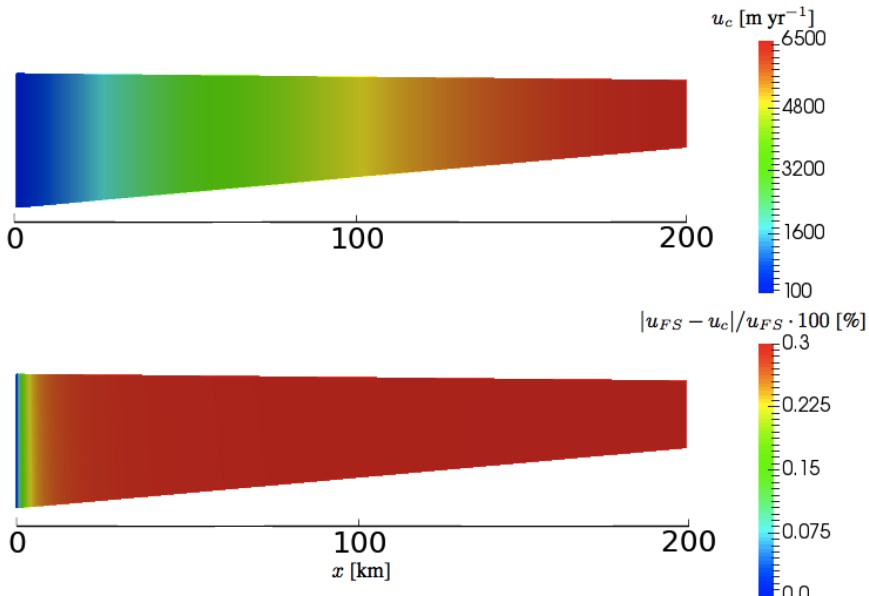

**Figure 2.** The horizontal velocity $u_c$ [m yr$^{-1}$] and node-wise difference $|u_{FS} - u_c|/u_{FS} \cdot 100$ [%] in the coupled solution for the 2D ice shelf ramp. The vertical scale is exaggerated 100 times. The ice thickness ranges from 400 to 200 m.

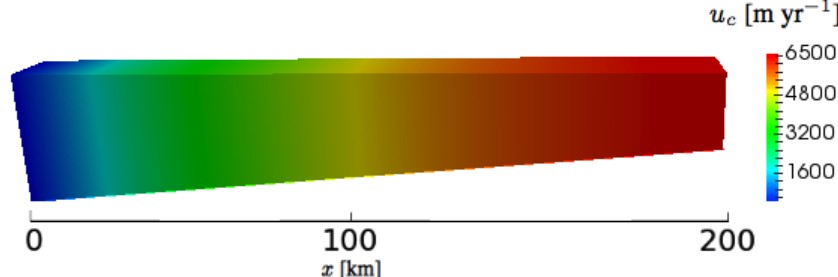

**Figure 3.** Horizontal velocity $u_c$ [ m yr$^{-1}$] from coupled model for the 3D ice shelf ramp with $x_c = (100, y)$ km.

### 4.2 Marine ice sheet

First, a diagnostic MIS experiment is performed in 2D to compare velocities for the initial geometry. After one time step, velocity differences between the coupled and FS models yield geometric differences. In prognostic experiments, velocity
5  differences can therefore be due to the coupling and to the different geometry for which the velocity is solved. Computation times for the FS and coupled model are presented for the prognostic case only.

### 4.2.1 Diagnostic MIS experiment

The domain starts with an ice divide at $x = 0$, where $u = 0$, and terminates at a calving front at $x = L = 1800$ km. An equidistant grid with grid spacing $\Delta x = 3.6$ km is used. Other values of constants and mesh characteristics are specified in Table A2. Gagliardini et al. (2016) showed that resolving grounding line dynamics with a FS model requires very high mesh resolution around the grounding line. However, Gladstone et al. (2017) showed that the friction law assumed in this study (see Sect. 2.2.1) reduces mesh sensitivity of the FS model compared to the Weertman friction law assumed in Gagliardini et al. (2016), allowing the coarse mesh used here. The bedrock [m] is negative below sea level and is given by

$$b(x) = 200 - 900\frac{x}{L}. \tag{21}$$

Basal melt is neglected and the surface accumulation $a_s$ [m yr$^{-1}$] is a function of the distance from the ice divide,

$$a_s(x) = \frac{\rho_w}{\rho}\frac{x}{L}. \tag{22}$$

This experimental setup is almost equivalent to Gladstone et al. (2017), except for that they applied a buttressing force to the FS equations. It is possible to parametrize buttressing for the SSA equations as well through applying a sliding coefficient (Gladstone et al., 2012b). This was not done here as it may introduce a difference between the FS and SSA models that is unrelated to the coupling.

The diagnostic experiments are run on a steady state geometry computed by the FS model. First, the experiment 'SPIN' in Gladstone et al. (2017) is performed, starting from a uniform slab of ice ($H$=300 m), applying the accumulation in Eq. (22) for 40 kyr, such that a steady state is reached. The geometry yielded from these SPIN runs (which include buttressing) is used in simulations without buttressing until a new steady state (defined as a relative ice volume change below $10^{-5}$) is reached. This removal of buttressing leads to grounding line retreat from 871.2 km to 730.8 km (Fig. 4).

Again, FS-only, SSA-only and the coupled model are applied to this setup. Where $u_{FS} \geq 5$ m yr$^{-1}$, the relative difference between $u_{FS}$ and $u_{SSA}$ is below 1.8%. The velocity $u_c$ is given in Fig. 4, with $d_{GL} = 30$ km such that 58% of the nodes in the horizontal footprint mesh are located inside $\Omega_{SSA}$ ($\theta = 0.58$). The coupled model converges after 27 FS iterations on the restricted domain $\Omega_{FS}$, compared to 24 Picard iterations in the FS model. The relative difference between $u_{FS}$ and $u_c$ is below 0.5% (Fig. 4), this small difference shows that $d_{GL} = 30$ km is sufficient. For this configuration, 4% of the FS nodes are located between $x_{GL}$ and $x_c$ with $d_{GL} = 30$ km, hence decreasing $d_{GL}$ does not affect the proportion of nodes in $\Omega_{FS}$ significantly. Therefore, $d_{GL}$ is kept equal to 30 km for the prognostic experiment.

### 4.2.2 Prognostic MIS experiment

The prognostic experiment is aimed to verify model reversibility as in Schoof (2007). Starting from the steady state geometry, the ice temperature $T$ is lowered over a period of 500 years from -10 °C to -30 °C and back according to

$$T(t) = -10(2 - \cos(2\pi t/500))°\text{C for } 0 \leq t \leq 500 \text{ yr}. \tag{23}$$

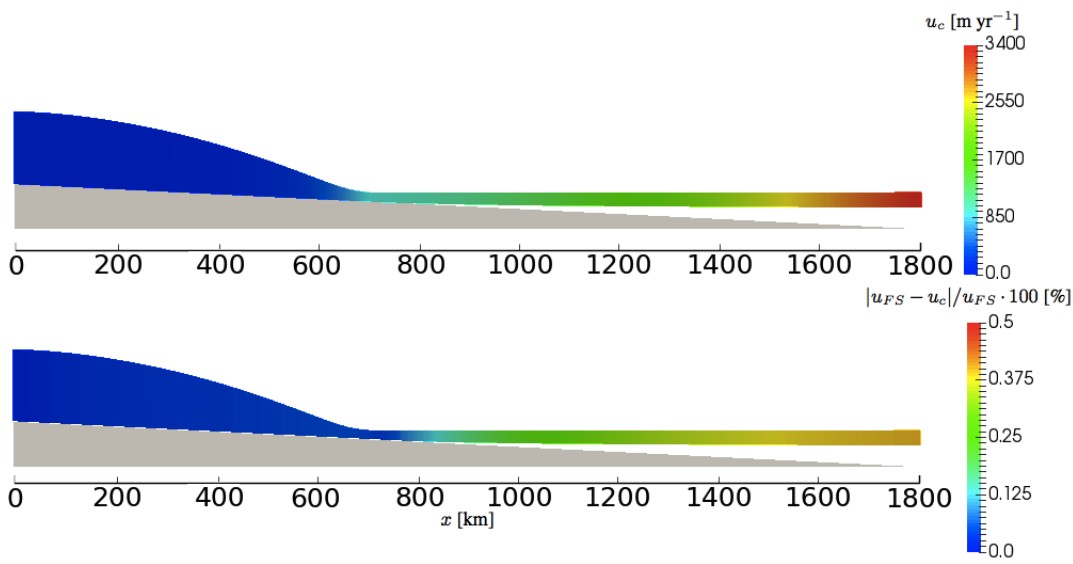

**Figure 4.** The coupled velocity $u_c$ [m yr$^{-1}$] and relative difference $|u_{FS} - u_c|/u_{FS} \cdot 100$ [%], for the diagnostic MIS experiment. The bedrock is shaded in grey, $x_{GL} = 730.8$ km, $x_c = 763.2$ km (the mesh resolution yields $||x_c - x_{GL}||_h$=32.4 km). The vertical scale is exaggerated 100 times with an ice thickness ranging from 1435 m to 296 m.

The resulting change in $\mathcal{A}$, see Eq. (4), induces a grounding line advance and retreat, and changes $\Omega_{SSA}$ by Eq. (17). Afterwards, $T = -10$ °C for 2500 years. Mass balance forcing is kept constant throughout. The length of one time step is 1 yr.

The maximum difference between $u_c$ and $u_{FS}$ after 3000 years is 10 m yr$^{-1}$, shown in Fig. 5, corresponding to a relative difference of 1.6%. The time evolution of $x_{GL}, u_b(x_{GL}), H(x_{GL})$ and the grounded volume $V_g$ are shown in Fig. 6 and Fig. 7. In general, $u_b$ is slightly higher in the coupled model, with a maximum difference of 5.3% in the entire experiment. The grounding line advances to $x_{GL} = 1036.8$ km in the FS model and $x_{GL} = 1044$ km in the coupled model. The FS model returns back to the original $x_{GL} = 730.8$ km, but the coupled model yields $x_{GL} = 734.4$ km, an offset of one grid point. The

maximum difference in thickness is 1%. After 3000 years, $V_g$ still decreases but the relative difference is below $10^{-5}$ between two time steps.

To investigate efficiency of the coupled model, the simulation is performed with ten different settings, where the maximum number of FS iterations per coupled iteration is varied from one to ten. Assembly of the FS matrix takes 75% of the computation time of the FS model (see $t_A$ in Table 1) and assembly time per FS iteration is similar for the coupled and FS model. Only 5%

of the computation time is used to solve the linearized FS system ($t_s$ in Table 1). For all coupled simulations, assemblying and solving the SSA matrix ($t_{SSA}$) takes 4-6%. All time that is left will be called overhead, $t_o$, which includes launching solvers, i.e. allocating memory space for vectors and matrices, the surface evolution and solvers for post processing. As expected, the total number of FS iterations is the smallest when just performing one FS Picard iteration per coupled iteration. However, the model then changes between solvers more often, meaning that both overheads and the time to solve the SSA model increase. It

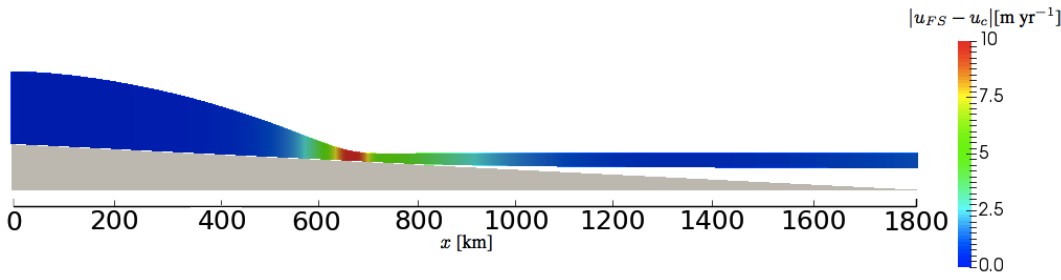

**Figure 5.** Absolute difference $|u_{FS} - u_c|$ [m yr$^{-1}$] after 3000 years. The vertical scale is exaggerated 100 times. The ice thickness ranges from 1445 m to 296 m.

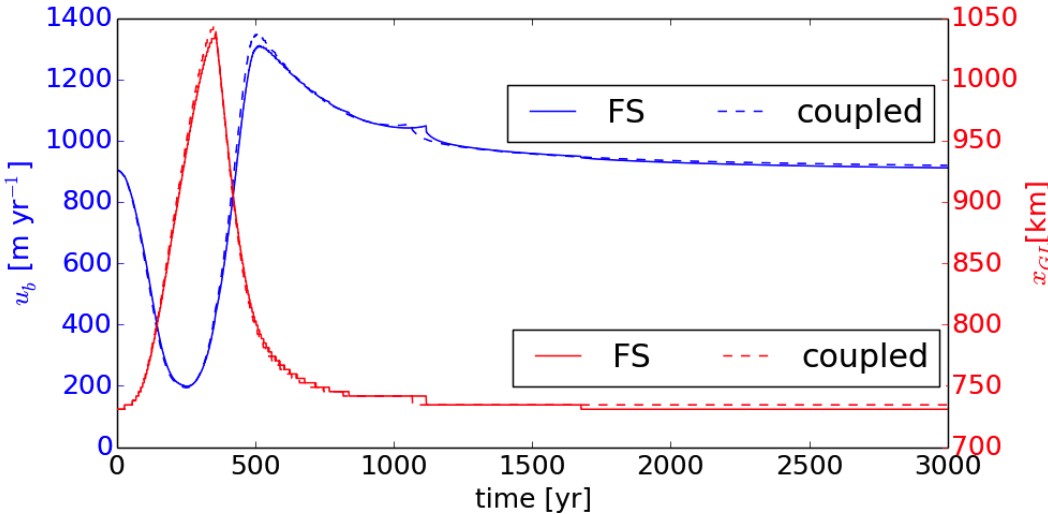

**Figure 6.** Time evolution of $x_{GL}$ (red) and $u_b(x_{GL})$ (blue) with solid lines for FS and dashed lines for the coupled model.

turns out that a limit of three FS Picard iterations per coupled iteration balances minimizing $t_o$ and $t_A$, yielding a 10% decrease of computation time with respect to the FS model. This speedup comes from a lower number of FS Picard iterations (Table 1) and a slight decrease of the time used to solve the linearized FS system (13% lower than the time that the FS model takes).

## 5   Discussion

The presented coupling is dynamic, since the coupling interface $x_c$ changes with grounding line changes, but the distance $d_{GL}$ that defines $x_c$ has to be chosen such that the FS velocity at the interface is almost independent of $z$. In the experiments described in Sect. 4, this is already the case at the grounding line. We propose that further studies let $\Omega_{SSA}$ be determined automatically, for example by a tolerance for the vertical variation of the horizontal velocities, which should be close to zero in

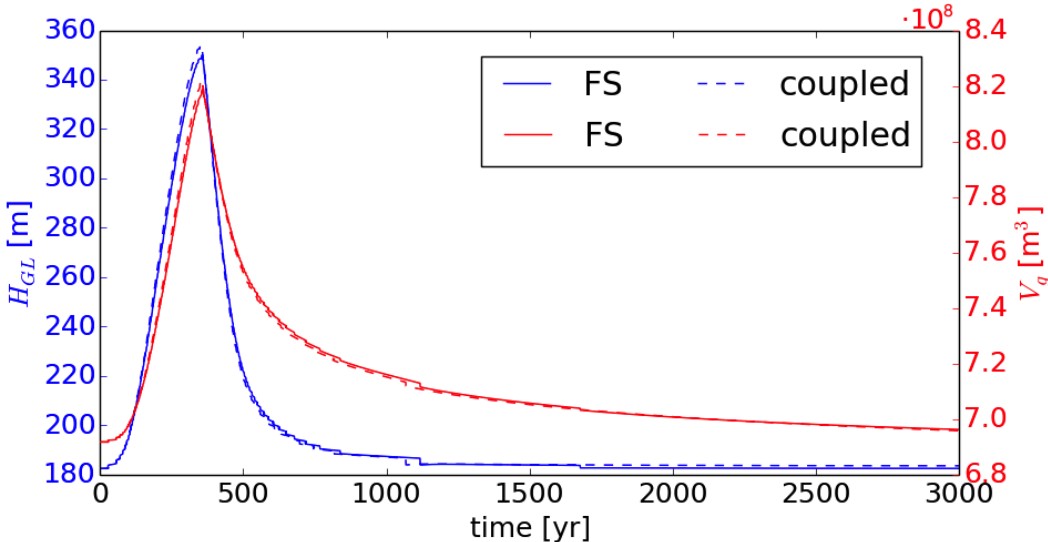

**Figure 7.** Time evolution of $H_{GL} = H(x_{GL})$ and grounded volume $V_g$ with solid lines for FS and dashed lines for the coupled model.

| model | $t_A$ [%] | $t_s$ [%] | # FS iter. | $t_o$ [%] | $t_{SSA}$ [%] | # coupled iter. | $t_{tot}$ [cpu s] |
|-------|-----------|-----------|------------|-----------|---------------|------------------|-------------------|
| FS    | 75        | 5         | 5.0        | 20        | -             | -                | 48641             |
| C10   | 68        | 4         | 4.6        | 25        | 4             | 2.7              | 49724             |
| C4    | 61        | 3         | 3.7        | 31        | 5             | 2.9              | 44143             |
| C3    | 59        | 3         | 3.6        | 33        | 5             | 3.1              | 44040             |
| C2    | 56        | 3         | 3.4        | 36        | 5             | 3.4              | 44334             |
| C1    | 49        | 3         | 3.2        | 43        | 6             | 4.2              | 47135             |

**Table 1.** Computation times for MIS simulation of 3000 yr with FS-only and coupled model. Model C$i$ denotes the coupled model with $i$ being the maximum number of non-linear FS iterations per coupled iteration, C5-C9 are omitted for brevity. The assembly time for $\mathbf{A}_{FS}$ is denoted $t_A$. All relative computation times are given in percentage of the total time $t_{tot}$. The number of FS and coupled iterations are averaged over the time steps.

order to allow for a smooth coupling to SSA. Another option is to use a posteriori error estimates based on the residual (Jouvet, 2016).

5     The current implementation in Elmer/Ice does not give as much speedup as expected from computation times of the FS- and SSA-only models for the ice shelf ramp ($t_{SSA} = 0.03 t_{FS}$) and from the performance estimates in Sect. 4.2. This is due to an inefficient matrix assembly. The assembly of the system matrix $\mathbf{A}_{FS}$ restricted to $\Omega_{FS}$ currently takes at least as much time as the assembly for the full domain $\Omega$, even though the domain $\Omega_{FS}$ is much smaller than $\Omega$; in Eq. (13), $\theta = 0.5$ for the ice shelf ramp and and $\theta = 0.58$ for the diagnostic MIS experiment. Since the assembly time dominates the total solution time in simple

2D simulations, this is problematic. The inefficient assembly is caused by the use of passive elements implemented in the overarching Elmer code (Råback et al., 2016), which allow de- and reactivation of elements. A passive element is not included in the global matrix assembly, but every element must be checked to determine if it is passive. The inefficient assembly can be overcome by implementing the coupling on a lower level, hardcoded inside the FS solver. This was done for the coupling of SIA and FS in Ice Sheet Coupled Approximation Levels (ISCAL, see Ahlkrona et al., 2016), which showed significant speedup when restricting the FS solver to a smaller domain. However, using passive elements is more flexible, since the coupling is independent of the solver used to compute velocities outside $\Omega_{SSA}$. One is free to choose between the two different FS solvers in Elmer/Ice (see Gagliardini et al., 2013) or to apply ISCAL. The latter is irrelevant in the experiments presented here since both the grounded and floating ice experiences low basal drag, and SIA is not capable of representing ice stream and shelf flow. Only a preliminary 3D experiment is performed here, since the current implementation is not sufficiently efficient to allow extensive testing in 3D. If the coupling is implemented efficiently such that the time spent on solving the FS equations on the restricted domain $\Omega_{FS}$ scales with the size of $\Omega_{FS}$, the computational work is expected to decrease significantly (see Sect. 4.2).

## 6   Conclusions

We have presented a novel FS-SSA coupling in Elmer/Ice, showing a large potential for reducing the computation time without losing accuracy. At the coupling interface, the FS velocity is applied as an inflow boundary condition to SSA. Together with the cryostatic pressure, a depth averaged contact force resulting from the SSA velocity is applied as a boundary condition for FS. The main finding of this study is that the two-way coupling is stable and converges to a velocity that is very similar to the FS model in the tests on conceptual marine ice sheets, and it yields a speedup in 3D.

In diagnostic runs, the relative difference in velocity obtained from the coupled model and the FS model is below 1.5% when applying SSA at least 30 km seaward from the grounding line. During a transient simulation, where the coupling interface changes dynamically with migration of the grounding line, the coupled model is very similar to the FS model, with a maximum difference of 5.3% in basal velocity at the grounding line. An offset of 3.6 km remains in the reversibility experiment in Sect. 4.3, which is within the range of the expected resolution dependence for FS models (Gladstone et al., 2017).

In experiments involving areas where SIA is applicable, this new FS-SSA model can be combined with the ISCAL method in Ahlkrona et al. (2016) that couples SIA and FS in Elmer/Ice. This mixed model is motivated by paleo-simulations, but reducing computational work by the combination of multiple approximation levels is also convenient for parameter studies, ensemble simulations, and inverse problems.

*Code availability.* The code of Elmer/Ice is available at https://github.com/ElmerCSC/elmerfem/tree/elmerice and can be redistributed and/or modified under the terms of the GNU General Public License as published by the Free Software Foundation; either version 2 of the License, or (at your option) any later version. An example of the coupling is provided at https://github.com/ElmerCSC/elmerfem/tree/elmerice/elmerice/Tests/MISMIP_FS-SSA, which is also linked to the doi 10.5281/zenodo.1202407. Besides that, it is possible to access the

version of the Elmer/Ice code including the coupling discussed in the paper, by using the SHA linked to the commit of the code, at https://github.com/ElmerCSC/elmerfem/archive/ba117583defafe98bb6fd1793c9c6f341c0c876.zip.

## Appendix A: Derivation of the interface boundary condition

The boundary condition in Sect. 3.1 between the FS and the SSA domains is derived following a standard procedure in FEM using the weak formulation of the equations. Let $\Omega_{FS} \in \mathbb{R}^d$, $d = 2, 3$, denote the open FS domain in two or three dimensions with the boundary $\Gamma_{FS}$. After multiplying Eq. (2) with a test function $v$ and integrating over the domain $\Omega_{FS}$, the weak form of Eq. (2) is

$$-\int_{\Omega_{FS}} v \cdot (\nabla \cdot \sigma) = \int_{\Omega_{FS}} \rho v \cdot g. \tag{A1}$$

Use the definition of $\sigma$ and the divergence theorem to rewrite Eq. (A1),

$$\int_{\Omega_{FS}} \eta \mathbf{D}(u) : \mathbf{D}(v) - \int_{\Omega_{FS}} p \nabla \cdot v = \int_{\Omega_{FS}} \rho v \cdot g + \int_{\Gamma_{FS}} v \cdot \sigma \cdot n. \tag{A2}$$

The operation $\mathbf{A} : \mathbf{B}$ denotes the sum $\sum_{i,j} A_{ij} B_{ij}$. The test function $v$ vanishes on the inflow boundary $\Gamma_i$, has a vanishing normal component on the bedrock boundary $\Gamma_b$, and lives in the Sobolev space $[W^{1,1/n+1}(\Omega_{FS})]^d$ (Jouvet, 2016), i.e.

$$v \in \mathcal{V}_0 = \{v \in [W^{1,1/n+1}(\Omega_{FS})]^d | \ v|_{\Gamma_i} = 0, \ v|_{\Gamma_b} \cdot n = 0\}. \tag{A3}$$

The space $\mathcal{V}_0$ has this form because the boundary conditions on $\Gamma_i$ and $\Gamma_b$ are of Dirichlet type. Furthermore, there is a lateral boundary $\Gamma_\ell$ for $\Omega_{FS} \in \mathbb{R}^3$, where the normal component also vanishes: $v|_{\Gamma_\ell} \cdot n = 0$ and we assume a vanishing Cauchy-stress vector for unset boundary conditions to velocity components, such that the integral over $\Gamma_\ell$ vanishes. Then, the boundary integral in Eq. (A2) consists of a sum of the remaining boundary terms

$$\int_{\Gamma_{FS}} v \cdot \sigma \cdot n = \sum_{i=1}^{d-1} \int_{\Gamma_b} f u \cdot t_i v \cdot t_i - \int_{\Gamma_w} p_w n \cdot v + \int_{\Gamma_{FSint}} v \cdot \sigma \cdot n, \tag{A4}$$

given by the boundary conditions on $\Gamma_b$ in Eq. (8) and (9), on the ocean boundary $\Gamma_w$ in Eq. (13), and the internal boundary $\Gamma_{FSint}$ between the FS and the SSA domains. The force $\sigma \cdot n$ on $\Gamma_{FSint}$ is determined by the SSA solution.

The open SSA domain $\Omega_{SSA} \in \mathbb{R}^2$, coupled to $\Omega_{FS} \in \mathbb{R}^3$, has the boundary $\Gamma_{SSA} = \Gamma_{SSAint} \cup \Gamma_{CF} \cup \Gamma_\ell$ where $\Gamma_{SSAint}$ is adjacent to $\Omega_{FS}$ and partly coinciding with $\Gamma_{FSint}$ (but of one dimension less) and $\Gamma_{CF}$ is at the calving front. Let $\mathbf{B}$ have the elements

$$B_{11} = 4\bar{\eta}\frac{\partial u}{\partial x} + 2\bar{\eta}\frac{\partial v}{\partial y}, \ B_{12} = B_{21} = \bar{\eta}\frac{\partial u}{\partial y} + \bar{\eta}\frac{\partial v}{\partial x}, \ B_{22} = 2\bar{\eta}\frac{\partial u}{\partial x} + 4\bar{\eta}\frac{\partial v}{\partial y}, \tag{A5}$$

when $d = 3$. If $d = 2$, then $B = 4\bar{\eta}\partial u/\partial x$. Then the SSA equations Eq. (6) can be written

$$\nabla_h \cdot \mathbf{B} = f_g, \tag{A6}$$

where $\boldsymbol{f}_g = \rho g H \nabla_h z_s$ and $\nabla_h$ is the horizontal gradient operator. The boundary condition on $\Gamma_{SSAint}$ is the Dirichlet condition, Eq. (20), and the force due to the water pressure at the calving front $\Gamma_{CF}$ is $\boldsymbol{f}_{CF}$, as in Eq. (13) but integrated over $z$. Define the two test spaces

$$\mathcal{W} = \{\boldsymbol{v} \in [W^{1,1/n+1}(\Omega_{SSA})]^{d-1} | \, \boldsymbol{v}|_{\Gamma_\ell} \cdot \boldsymbol{n} = 0\}, \quad \mathcal{W}_0 = \{\boldsymbol{v} \in \mathcal{W} | \, \boldsymbol{v}|_{\Gamma_{SSAint}} = 0\}. \tag{A7}$$

Multiply Eq. (A6) by $\boldsymbol{v} \in \mathcal{W}_0$ and integrate. The weak form of Eq. (A6) is

$$\int_{\Omega_{SSA}} \boldsymbol{v} \cdot (\nabla_h \cdot \mathbf{B}) = \int_{\Omega_{SSA}} \boldsymbol{v} \cdot \boldsymbol{f}_g. \tag{A8}$$

Apply the divergence theorem to Eq. (A8) to obtain

$$-\int_{\Omega_{SSA}} \nabla_h \boldsymbol{v} : \mathbf{B} + \int_{\Gamma_{SSA}} \boldsymbol{v} \cdot \mathbf{B} \cdot \boldsymbol{n} = -\int_{\Omega_{SSA}} \nabla_h \boldsymbol{v} : \mathbf{B} + + \int_{\Gamma_{CF}} \boldsymbol{v} \cdot \boldsymbol{f}_{CF} + \int_{\Gamma_{SSAint}} \boldsymbol{v} \cdot \boldsymbol{f}_{SSA} = \int_{\Omega_{SSA}} \boldsymbol{v} \cdot \boldsymbol{f}_g. \tag{A9}$$

A mesh is constructed to cover $\Omega_{FS}$ and $\Omega_{SSA}$ with nodes at $\boldsymbol{x}_i$. In the finite element solution of Eq. (A9), the linear test function $\boldsymbol{v}_i \in \mathcal{W}_0$ is non-zero at $\boldsymbol{x}_i$ and zero in all other nodes. The integral over $\Gamma_{SSAint}$ vanishes when $\boldsymbol{v} \in \mathcal{W}_0$. The finite element solution $\boldsymbol{u}_h$ of Eq. (A6) and (A9) satisfies

$$-\int_{\Omega_{SSA}} \nabla_h \boldsymbol{v}_i : \mathbf{B}(\boldsymbol{u}_h) + \int_{\Gamma_{CF}} \boldsymbol{v}_i \cdot \boldsymbol{f}_{CF} - \int_{\Omega_{SSA}} \boldsymbol{v}_i \cdot \boldsymbol{f}_g = 0, \; \boldsymbol{x}_i \in \Omega_{SSA} \cup \Gamma_{CF}. \tag{A10}$$

It follows from Eq. (A9) that with a test function $\boldsymbol{v}_i \in \mathcal{W}$ that is non-zero on $\Gamma_{SSAint}$ and the solution $\boldsymbol{u}_h$ from Eq. (A10)

$$\int_{\Gamma_{SSAint}} \boldsymbol{v}_i \cdot \boldsymbol{f}_{SSA} = \tag{A11}$$

$$= \int_{\Omega_{SSA}} \nabla_h \boldsymbol{v}_i : \mathbf{B}(\boldsymbol{u}_h) - \int_{\Gamma_{CF}} \boldsymbol{v}_i \cdot \boldsymbol{f}_{CF} + \int_{\Omega_{SSA}} \boldsymbol{v}_i \cdot \boldsymbol{f}_g, \tag{A12}$$

$$\boldsymbol{x}_i \in \Omega_{SSA} \cup \Gamma_{CF} \cup \Gamma_{SSAint}. \tag{A13}$$

The first integral in Eq. (A12) corresponds to $(\mathbf{A}_{SSA}\boldsymbol{u}_{SSA})_i$ in Sect. 3.1 and $b_{SSAi}$ to the second and third integrals. By Eq. (A10), the right hand side of Eq. (A12) vanishes for all $\boldsymbol{x}_i$ in $\Omega_{SSA}$ and on $\Gamma_{CF}$, but for a node on the internal boundary, $\boldsymbol{x}_i \in \Gamma_{SSAint}$, the force $\boldsymbol{f}_{SSA}$ from the ice due to the state $\boldsymbol{u}_h$ in $\Omega_{SSA}$ is obtained. The internal pressure in the ice in $\Omega_{SSA}$ is assumed to be cryostatic as in Eq. (18). The total force on $\Gamma_{FSint}$ consists of one component due to the state $\boldsymbol{u}_h$ at $\Gamma_{SSAint}$ and one due to the cryostatic pressure there. Let $\Omega^*_{SSA}$ denote the mesh on $\Omega_{SSA}$ which is extruded in the $z$-direction. The common boundary between $\Omega_{FS}$ and $\Omega^*_{SSA}$ is $\Gamma_{FSint}$ and let $\boldsymbol{f}^*_{SSA}$ be the stress force there, independent of $z$. Since $\int_{z_b}^{z_s} \boldsymbol{f}^*_{SSA} = \boldsymbol{f}_{SSA}$ at $\Gamma_{FSint}$, we have $\boldsymbol{f}^*_{SSA} = H^{-1}\boldsymbol{f}_{SSA}$. Let $\boldsymbol{v}_i$ be a test function on $\Omega_{FS} \bigcup \Omega^*_{SSA}$ which is non-zero on $\Gamma_{FSint}$ and zero in all other nodes. Then the weak form of the force balance at $\Gamma_{FSint}$ is

$$\int_{FSint} \boldsymbol{v}_i \cdot \boldsymbol{\sigma} \cdot \boldsymbol{n} = \int_{FSint} \boldsymbol{f}^*_{SSA} \cdot \boldsymbol{v}_i - \int_{FSint} \rho g(z_s - z)\boldsymbol{n} \cdot \boldsymbol{v}_i = \int_{FSint} H^{-1}\boldsymbol{f}_{SSA} \cdot \boldsymbol{v}_i - \int_{FSint} \rho g(z_s - z)\boldsymbol{n} \cdot \boldsymbol{v}_i, \tag{A14}$$

and the corresponding strong form of the boundary condition at $\Gamma_{FSint}$ is

$$\boldsymbol{\sigma} \cdot \boldsymbol{n} = H^{-1}\boldsymbol{f}_{SSA} - \rho g(z_s - z)\boldsymbol{n}, \tag{A15}$$

cf. Eq. (19). Thus, by computing the residual as in Eq. (19) the two finite element solutions in $\Omega_{FS}$ and $\Omega_{SSA}$ are coupled
together at the common boundary $\Gamma_{FSint}$ and $\Gamma_{SSAint}$.

*Author contributions.* NK, EvD, RvdW, MvG, PL, LvS designed the study. EvD implemented the coupling and carried out the numerical simulations, with support from TZ and CG. EvD drafted the manuscript with support from NK, and all authors contributed to the final version.

*Competing interests.* All other authors declare that they have no conflict of interest.

*Acknowledgements.* This work has been supported by FORMAS grant 214-2013-1600 to Nina Kirchner. Thomas Zwinger's contribution was supported by the Academy of Finland (grant number 286587). The computations were performed on resources provided by the Swedish National Infrastructure for Computing (SNIC) at PDC Center for High Performance Computing at KTH. We are grateful to Mika Malinen, Peter Råback and Juha Ruokolainen for advice in developing the coupling, to Rupert Gladstone for providing the setup as in Gladstone et al. (2017) and to Felicity Holmes, Guillaume Jouvet and Daniel Farinotti for their feedback on a draft of the manuscript. We wish to acknowledge the constructive comments of two anonymous reviewers, which contributed to improve the manuscript.

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

**Table A1.** Numerical values of the constants used in the ice shelf ramp experiment. Since the shelf is afloat, there is no sliding at the base.

| Parameter | Symbol | Value | Unit |
|---|---|---|---|
| Ice density | $\rho$ | 900 | $\text{kg m}^{-3}$ |
| Water density | $\rho_w$ | 1000 | $\text{kg m}^{-3}$ |
| Gravitational acceleration | $g$ | 9.81 | $\text{m s}^{-2}$ |
| Fluidity parameter | $\mathcal{A}$ | $10^{-16}$ | $\text{Pa}^{-3}\,\text{yr}^{-1}$ |
| Number elements | $N_z$ | 10 | |
| | $N_x$ | 120 | |
| | $N_y$ | 10 | |
| Picard convergence tolerance | $\varepsilon_P$ | $10^{-3}$ | |
| Coupled convergence tolerance | $\varepsilon_c$ | $10^{-4}$ | |

**Table A2.** Numerical values of the constants for the MIS experiment.

| Parameter | Symbol | Value | Unit |
|---|---|---|---|
| Ice density | $\rho$ | 910 | kg m$^{-3}$ |
| Water density | $\rho_w$ | 1000 | kg m$^{-3}$ |
| Gravitational acceleration | $g$ | 9.81 | m s$^{-2}$ |
| Sliding parameter | $\beta$ | $7 \cdot 10^{-6}$ | MPa m$^{-4/3}$ yr$^{1/3}$ |
| Temperature | $T$ | -10 | $^\circ$ C |
| Number elements | $N_z$ | 11 | |
| | $N_x$ | 500 | |
| Picard convergence tolerance | $\varepsilon_P$ | $10^{-4}$ | |
| Coupled convergence tolerance | $\varepsilon_c$ | $10^{-4}$ | |
| Time step | dt | 1 | yr |