# Peer review of "Dynamically coupling Full Stokes and Shallow Shelf Approximation for marine ice sheet flow using Elmer/Ice (v8.3)"

_Geoscientific Model Development, 2017_

## Short Comment (SC1) · 12 Mar 2018

The precise version of the code discussed in the manuscript must be made available. The current best practice is for this code to be uploaded to a public repository and a DOI assigned. The DOI should be cited in the manuscript. github is inadequate because it does not readily link to the precise version of the code. However, making github code citable is not difficult; see: https://guides.github.com/activities/citable-code/

---

## Author Comment (AC1) · 14 Mar 2018

We agree with J. C. Hargreaves that the version of the code discussed in the manuscript must be made available. However, assigning a DOI to the code would require a release of the underlying multi-physics code Elmer to which we contributed. Since new releases of Elmer are created approximately once a year, this is not desirable. Instead of a DOI, we can provide a SHA which is linked to a specific commit. In this case, the SHA linked to the commit is https://github.com/ElmerCSC/elmerfem/commit/ba117583defafe98bb6fd1793c9c6f341c0c876e. The combination of the GitHub directory to download from and the SHA is unique and

thus provides the version of the code discussed in the manuscript. Therefore, we will add the SHA to the code availability section of the paper.

---

## Short Comment (SC2) · 15 Mar 2018

The link provided does not resolve to the bundle of code that was developed in the manuscript. It is merely a "corrected version of MISMIP_FA-SSA test SIF".

You clearly understand that there must be a unique version number or identifier of some kind that relates to your code. Linking to Zenodo or similar, and providing a DOI is not anymore a public release of Elmer than is the github development directory. It is simply a robust citable link to the code developed in the paper.

If you can provide the link here in the open review that will be appreciated, as it is

[Discussion paper]

[Figure]

expected that editor and reviewers will want to look at the code (although - do not worry - there is no formal review of the code!).

---

## Author Comment (AC2) · 19 Mar 2018

We apologize for not providing detailed instructions on accessing the code in our previous reply. It is possible to access the version of the code discussed in the paper, linked to the unique SHA, by using the following git command, after having cloned the Elmer repository:

git checkout ba117583defafe98bb6fd1793c9c6f341c0c876

or download the repository from

https://github.com/ElmerCSC/elmerfem/archive/ba117583defafe98bb6fd1793c9c6f341c0c876.zip

[Figure]

To ensure robustness and longevity of the link to the code discussed in the paper, the code provided at https://github.com/ElmerCSC/elmerfem/tree/devel/elmerice/Tests/MISMIP_FS-SSA is now also linked to the DOI https://doi.org/10.5281/zenodo.1202407, which will be added to the code availability section of the paper.
* * *

---

## Referee Comment (RC1) · Anonymous Referee #1 · 29 Mar 2018

**1   Summary:**

This manuscript proposes a new method to couple the Full Stokes and Shelfy Stream Approximations so that different parts of a model domain can rely on different approximations of the stress balance equations. The idea of combining different stress balance approximations has been around for some time, with limited success concerning the inclusion of the Full Stokes equations, so it is great to see a new method being proposed. After the description of the method, several diagnostic and prognostic examples are shown to assess the accuracy of the solution and the gain of the coupling in terms of computational time.

[Figure]

There are several points either unclear or missing in the manuscript that are detailed below, including two critical ones that preclude me from accurately assessing this new method in the current version of this manuscript. The first one is that it is really difficult to follow the derivation of the coupling, because the equations are hard to follow. I understand that this is a very technical problem, but the goal is to describe in a way that is accessible to most interested readers, so some clarifications are needed. The second one is that there is no example showing the impact of the coupling method in a case where the Full Stokes and Shelfy Stream Approximations exhibit a significantly different behavior. This might be the case for the marine ice sheet experiment, but results from the Shelfy Stream Approximations are never shown, so it is impossible to undoubtedly assess the capability of this new coupling method.

**2   Major comments:**

The main point of this paper is to describe a new coupling method between the Full Stokes and Shelfy Stream Approximation equations. Unfortunately the current version of the manuscript is written in a way that makes understanding this new method quite challenging. First of all, all the derivation is hidden in the appendix, while it should be the core of the paper. Second, the appendix is a list of equations with no clear path to follow the demonstration, often jumping from one equation to the next with no explanation. Specifically, there should be a few sentences at the beginning explaining the method used to derive the additional force applied at the boundary between the two subdomains. It should be clearly stated when new terms are introduced and where they come from, or if it is a new definition (Let us define $X$ as ...). I tried to understand the derivation of the force, but I must say that I am still quite confused by several parts despite spending ample time on it.

I find the introduction a bit biased to justify the need of this new coupled model. Having

the opportunity to combine different stress balance approximations is indeed a genuine idea worth pursuing, and worth some investigations. So there is no need to emphasize the importance of solving a Full Stokes contact problem at the grounding line when recent studies suggest a limited impact (Pattyn and others, 2013), or to advertise using friction laws that require limited resolution around the grounding line (Gladstone and others, 2013) because it is easier to do with a Full Stokes model that would otherwise require too much computational capabilities to be solved at very high resolution around the grounding line. I would like to see the introduction a bit more in line with the literature, which would not diminish the importance of this manuscript.

The manuscript details the coupling between the Full Stokes and Shelfy Stream Approximations, but there are no details about the other difficulties caused by this coupling. Especially, there is no detail on how the domain is discretized into a 2D and a 3D part, and how this division evolves with time (how is the distance to the grounding line computed? how are elements switched between 2D and 3D?). There is also no detail on how the surface evolution is connected into the two parts of the domain. In one part of the domain, only one equation describing the thickness evolution needs to be solved, while in the other part, two equations describing the evolution of the upper and lower surface elevations need to be solved. Similar to the stress balance equation, some explanations must be added to explain the coupling in the surface evolution equations.

The notations in the equations are not always consistent, for example between eq.(4) and eq.(6). They both describe a similar quantity but one is based on the components of the tensor, while the other one is based on the velocity derivatives, for no obvious reasons. Same for $\eta$ and $\bar{\eta}$ in eq.(3) and eq.(5). They both depend on the velocity, but in one case the dependence is explicitly stated while it is not in the other case, which tends to be confusing. There are also many terms introduced that are not necessary, adding more confusion.

Results for the marine ice sheet experiment with a pure Shelfy Stream Approximation

should be added to see how different the solution is from a Full Stokes model. The objective here is to assess the algorithm in a case where the Shelfy Stream Approximation and Full Stokes solutions are different. We don't know here if this experiment leads to different results with the two stress balance equations, and therefore there is no evidence that the coupling works in a case where the two solutions are different. So until we see how the coupling works in a case where the Shelfy Stream Approximation and Full Stokes equations lead to significantly different solutions, it is not possible to assess the capability of this coupling method to correctly produce an accurate solution that needs Full Stokes on a part of the domain.

**3  Technical suggestions:**

Note that the line numbering on each page starts with a different number, so I did my best to be clear but it might sometimes induce some confusion.

p.1 l.2: "their non-linearlity" → "the non-linearity"

p.1 l.2: "are used" → "are commonly used"

p.1 l.8: "periodical temperature" → "periodic temperature"

p.1 l.10: "modeling an ice sheet complex" → "modeling a complex ice sheet"

p.1 l.15: increased attention to what? (missing words)

p.1 l.16: Quantify "much". Also it seems that at least in Greenland, the majority of the changes are caused by surface mass balance changes (Enderlin and others, 2014).

p.2 l.1: remove "strongly": some materials rheology are much more non-linear that ice.

p.2 l.7: add Hindmarsh (2004) as a reference for the hybrid models

p.2 l.13: I disagree with the interpretation of the Pattyn and others (2013) results. In my

opinion they show that models including membrane stress and whose grid resolution is sufficiently small capture grounding line evolution in a relatively similar way.

p.2 l.15: "complexes" → "systems"

p.2 l.17: add a reference for the Ice Sheet System Model

p.2 l.23-25: The important question is not so much if friction laws depending on the effective pressure law are faster, but to figure out which ones are more accurate and a allow a good description of the bedrock underlying the ice.

p.3 l.7: "strain rate" → "strain rate tensor"

p.3 l.15: remove "highly"

p.3 l.16: I don't see the link between the non-linearity of the Full Stokes model and the derivation of simplified approximations. Most the approximations are also non-linear, and the main purpose of these approximations is to solve a system with less than four coupled unknowns, as is the case with Full Stokes problem.

p.3 l.20: So what terms are neglected in the Shallow Ice Approximation?

p.4 l.2: How is $p$ eliminated?

p.4 l.2: I think the main difference that should be explained is that in the Full Stokes case, one has to solve a 3D problem with 4 unknowns, while in the Shallow Shelf Approximation, one has to solve a 2D problem with 2 unknowns.

Fig.1 caption: What is $d_{GL}$? Also change "coupling interface" and "grounding line" by "the coupling interface" and "the grounding line"

p.4 l.5: "parametrized" → "represented"

p.4 Eq.(9): $z^\star$ does not seem to depend on $N$ in eq.(10).

p.5 l.14: "volume gain" is confusing, rephrase.

p.5 l.18: Why introduce the ice flux here. This is a new quantity that is never used in the paper, and could be simply replaced by its description ($H\bar{u}, H\bar{v}$)

p.5 l.20-23: This 2D/3D explanations are a bit confusing. It is not clear if the SSA part of the model is still in 2D or 3D but the equations are solved only in 1D or 2D on a layer of the model, or if the mesh is indeed changed to 1D or 2D for the SSA, in which case it is not accurate to refer to 2D and 3D models only, and it would be more accurate to say 1D/2D and 2D/3D. Also, there are no details on what is done for the different parts of the mesh.

p.6 l.1-5: As mentioned just above, we don't have any detail on how the discretization of the problem is done in the two parts, and how they are connected. For example, how is the domain decomposed into 2D and 3D (or 1D and 2D), and how does this evolve with time. Details need to be added to understand this part, especially the connection between the two parts of the domain.

p.6 l.12: How accurate is the residual free bubbles method?

p.7 l.3: "the SSA equations": maybe add here that they are used as Dirichlet boundary conditions right away, that will be more clear.

p.7 l.6: What are $A$ and $b$? Define them.

p.7 l.8: What is $A_{SSA}$? Define it.

p.7 l.8: "without the Dirichlet conditions": this is really confusing. It is really hard to follow what is going on here, as none of the terms are explained or detailed.

p.7 l.10: see Appendix A: the derivation of the force applied at the interface of the Full Stokes and Shelfy Stream parts of the domain is the core of the paper. This explanation of where this force comes from and how it was derived need to be entirely moved to the main manuscript, and not hidden in an Appendix.

p.7 l.12: Why not take the depth-averaged velocity? This would be more consistent

with the Shallow Shelf Approximation that computes a vertically integrated solution.

p.7 l.21: The surface evolution is solved differently for the two parts of the domain, but no detail is provided on how these two parts are connected, how the equations are actually solved (together or one after the other), and what is the impact in terms of continuity of the surface elevations, or feedback between the two parts.

p.8 l.29: How are the free surface equations solved to ensure continuity of the solution between the two parts of the domain?

p.10 Fig.3: Consider using round up the numbers in the colorbar. Also "where $x_c$" → "with $x_c$"

p.11 l.24: What are the basal conditions applied for this set-up?

p.11 l.25: Describe the "SPIN" experiment.

p.11 l.11 "minimal" → "small/limited"

p.11 l.11: "For the used mesh resolution" → "For this configuration"

p.11 l.13: "equal for" → "equal to 30 km for"

p.11 l.15: Add that the temperature is varied for 500 years and then kept constant for the remaining 2500 years. The equation of the temperature and its description look quite contradictory.

p.12: results for both diagnostic and prognostic marine ice sheet experiments with a pure SSA model should be added for comparison.

p.13 Fig.6 and Fig.7 captions: "solid line" and "dashed line" → "solid lines" and "dashed lines"

p.14 l.3: "follows" → "comes"

p.14 l.16-18: For the prognostic experiment, how is the repartition between the two subdomains computed and set-up? Especially, how is the distance to the grounding

line for each point computed (in 3D), and how is the mesh changed from 2D to 3D and vice versa when the grounding line evolves. For elements that are switched from the Shallow Shelf Approximation to the Full Stokes approximation, what is used for the velocity at the beginning of the step, especially the vertical velocity?

p.14 l.5: What is $A_{FS}$? Define it.

p.14 l.7: "and and"

p.14 l.19: "the computational work will decrease significantly". This is quite speculative and should be at least replaced by "is expected to"

p.15 l.11: "multiplying with": multiplying what?

p.16 l.24: "partly coinciding": why partly? There is only one interface between the two subdomains.

p.16 l.11: Eq.(A6) and Eq.(A9) are the same but just rearranged. $u_h$ is not solution of both as it is the same one.

p.16 l.14: What is $f_{SSA}$? Is that how you define it: $f_{SSA} = A\dot{n}$? In the case why is the first integral over $\Gamma_{SSAint}$ and the second one over $\Gamma_{SSA}$? If not, what is $f_{SSA}$?

p.17: 20: What about the across flow direction?

p.17 l.22: I don't understand where it comes from. You are trying to estimate the last term of Eq.(A4) to apply this force to the Full Stokes part of the domain. The force applied by one subdomain on the other and vice versa are equal, so instead you try to estimate the second term in eq.(A11). But is it ok to have $A$ instead of $\sigma$? And how do you get from the weak form to the local equation This sounds pretty abrupt.

p.21: There is no mention in the paper of what friction law is applied.

**4 Bibliography:**

Enderlin, Howat, Jeong, Noh, van Angelen, and van den Broeke, An improved mass budget for the Greenland ice sheet, Geophys. Res. Lett., 2014.

Gladstone, Warner, Galton-Fenzi, Gagliardini, Zwinger, and Greve, Marine ice sheet model performance depends on basal sliding physics and sub-shelf melting, The Cryosphere, 2017.

Hindmarsh, A numerical comparison of approximations to the Stokes equations used in ice sheet and glacier modeling, J. Geophys. Res., 2004.

Pattyn, and others, Grounding-line migration in plan-view marine ice-sheet models: results of the ice2sea MISMIP3d intercomparison, J. Glaciol., 2013.

---

## Referee Comment (RC2) · Anonymous Referee #2 · 19 May 2018

This work makes modifications to a well-known and widely used glacial flow model Elmer-Ice to allow an approximation to the nonlinear Stokes problem – the Shallow Shelf Approximation – to be solved for a large proportion of the floating part of the domain, in such a way that the two solves are consistent with respect to the force balance. The significance of such a modification is the expense involved with the stokes solve. Not only does it have more degrees of freedom, but it leads to a problem which in the variational sense is a saddle-point and not a minimization problem, leading to difficulty in discretisation; and, though it is not flagged by the authors, there is an extra complication in solving for ice shelves, as a floatation condition cannot be assumed, allowing for oscillation and instability in the vertical momentum condition that must be

artificially damped. The work is a nice follow-on slash complement to two other works in the literature: Seroussi et al 2012, which couples together different approximations to the FS problem in a diagnostic setting; and Ahlkrona et al 2016, which dynamically couples FS to SIA in an evolving ice sheet, and is deserving of publication.

i only have 2 general comments:

1) My one general comment is that the underlying premise seems to be that this method will reduce computational resource requirements. The reduction in the test cases seems to be low; but this is explained in the discussion, and I am not presently commenting on this. It is rather that presumably this coupling is meant to be applied to regional and continental scale simulations of marine ice sheets. As SSA is only solved on the shelf, the savings are limited by the portion of the domain covered by ice shelves. Quick googling tells me that ice shelves currently represent 10% of Antarctic total area – significantly less than in the test simulations in the paper. So how much more efficient is such a coupling meant to be compared to FS only continental simulations? But I am curious about the issue that i bring up in my first paragraph, but was not addressed. Durand et al 2009 discusses their approach to prevent instability in the vertical due to ice shelves not being in floatation (they added a degree of implicitness to the velocity solver – their eqn 15). I have wondered if this potential instability affects the solution of FS for a marine ice sheet, and possibly artificially enforces archimedean floatation. I would be curious to know about how your coupling affects this issue ,i.e. the need for the "fix" – could you remove the fix for your coupled experiments?

2) There is not much discussion on how the FS and SSA domains are updated dynamically (node reassignment etc). Although you touch on it in the discussion, i would like to see a subsection in the methods section briefly detailing this, even if it makes use of already-existing frameworks. Apologies if such text is there and i overlooked it.

Specific comments:

p2 l 10: MISMIP3D

p3 line 7: strain rate tensor

p3 l 9: here they are just the Stokes equations. ("Full Stokes" arose because glaciologists realised the equations they had been solving for years were an approximation to Stokes with power-law rheology)

eq 4: your notation for the the 2nd invariant of tensors (in this case D) is a bit subtle, you might consider something else – this is simply a suggestion though

p3 l 22: "h represents the horizontal components" is a bit ambiguous, and you do not use the subscript in eq (6)

eq (9): is z* a function of effective pressure, which is not defined, or of H as it appears to explicitly be from eq (10)?

Section 2.3, abbreviations of two dimensions and three dimensions seem awkward, and be clear by 2D you mean the x-z plane.

p 5 l 22: say what 3 variables are for FS in 2D.

p 5 l 23: say what you mean by stokes being a saddle-point problem as many readers will not know what this means.

p 5 l 23: "with all its consequences for numerical treatment" – give examples of these consequences, with references. Most readers will not be familiar with this literature, and others (like me) will be only familiar with some of it.

p 6 l 25: just \theta, not \theta Nz

p6 l 26: say this is approximate, as it must be due to boundaries etc.

p6 l 6: by "number of unknown variables" do you mean degrees of freedom; or (u,v,w,p) i.e. 4 versus (u,v) i.e. 2? You are saying that, per node, the former takes twice as long to assemble in a matrix versus the latter? I find this difficult to believe in general. What about differing orders of polynomial in the basis functions, and complexity of interaction

between DoFs of different variables?

p 6 l 16: is there a theoretical basis for d_{GL} – such as the estimate of the non-hydrostatic boundary layer from Schoof (2011, JFM)? Or arbitrary?

p 7 eq (17): this is an initial guess for the membrane stress at the interface, yes? Would a better one not be

\sigma_{FS} \cdot \vec{n} = \rho_w g (-z) if z<0,

i.e. the condition which would be applied if xc was a calving front? (This would be exact in the 2D case with no buttressing seaward of x_c; and i wonder if it would reduce iteration count in general)

p 9 l 12: "assembly time ... almost doubles" – i think this is the issue you address in the discussion? would be good to say that it is addressed in the discussion, as i was confused by this when i read it.

p 9 l 15: is v velocity in the y-direction? better say so. aside from no normal flow, what is the other BC at the lateral boundary? no stress? no flow?

p 11 line 8: 58% of the nodes – you mean in the projection of the grid to the x-y plane?

p 12 paragraph at line 13: found this discussion a bit difficult to follow – is there any way the main performance points can be summarised in a table? also limited –> varied

p 14 discussion at line 10: if you were to make this change re: node assignment at a lower level, how would it affect the ability to easily update the Full Stokes sub-domain?

p 16 line 24: you previously used A as a symbol for a matrix

p 16 line 4: did you define f_{CF}?

p 16 eq (A7) and below: suggest to use w as a test function symbol

p 16 eq (A9) the 1st boundary integral is over \Gamma_{SSA} not \Gamma_{SSAint}

---

## Author Comment (AC3) · 18 Jun 2018

Anonymous Referee #1 comments and response

We would like to thank the reviewer very much for the time and effort put into reviewing our manuscript. A point-by-point reply (in blue) to each comment (in black) by Referee #1 are given below. Specific changes in the manuscript are written in blue italics.

Summary:
This manuscript proposes a new method to couple the Full Stokes and Shelfy Stream Approximations so that different parts of a model domain can rely on different approximations of the stress balance equations. The idea of combining different stress balance approximations has been around for some time, with limited success concerning the inclusion of the Full Stokes equations, so it is great to see a new method being proposed. After the description of the method, several diagnostic and prognostic examples are shown to assess the accuracy of the solution and the gain of the coupling in terms of computational time.

There are several points either unclear or missing in the manuscript that are detailed below, including two critical ones that preclude me from accurately assessing this new method in the current version of this manuscript. The first one is that it is really difficult to follow the derivation of the coupling, because the equations are hard to follow. I understand that this is a very technical problem, but the goal is to describe in a way that is accessible to most interested readers, so some clarifications are needed. The second one is that there is no example showing the impact of the coupling method in a case where the Full Stokes and Shelfy Stream Approximations exhibit a significantly different behavior. This might be the case for the marine ice sheet experiment, but results from the Shelfy Stream Approximations are never shown, so it is impossible to undoubtedly assess the capability of this new coupling method.

Major comments:
The main point of this paper is to describe a new coupling method between the Full Stokes and Shelfy Stream Approximation equations. Unfortunately the current version of the manuscript is written in a way that makes understanding this new method quite challenging. First of all, all the derivation is hidden in the appendix, while it should be the core of the paper. Second, the appendix is a list of equations with no clear path to follow the demonstration, often jumping from one equation to the next with no explanation. Specifically, there should be a few sentences at the beginning explaining the method used to derive the additional force applied at the boundary between the two subdomains. It should be clearly stated when new terms are introduced and where they come from, or if it is a new definition (Let us define X as ...). I tried to understand the derivation of the force, but I must say that I am still quite confused by several parts despite spending ample time on it.

We agree that the manuscript improved when introducing terms and equations more clearly. Specifically, we added information about the additional force, as explained in response to the technical suggestions for p.7 l.8 and l.10, p. 16 and p. 17.
Besides that, we have rewritten the introduction such that it is clearer that the FS-SSA coupling proposed is more a domain decomposition method than actually coupling the stresses themselves. Specific comments are addressed below (suggestions to p. 16-17).

I find the introduction a bit biased to justify the need of this new coupled model. Having the opportunity to combine different stress balance approximations is indeed a genuine idea worth pursuing, and worth some investigations. So there is no need to emphasize the importance of solving a Full Stokes contact problem at the grounding line when

recent studies suggest a limited impact (Pattyn and others, 2013), or to advertise using friction laws that require limited resolution around the grounding line (Gladstone and others, 2013) because it is easier to do with a Full Stokes model that would otherwise require too much computational capabilities to be solved at very high resolution around the grounding line. I would like to see the introduction a bit more in line with the literature, which would not diminish the importance of this manuscript.

Thank you for acknowledging the importance of this manuscript. The interpretation of Pattyn and others (2013) is addressed in the technical suggestion p.2 l.13. The section on a friction law that requires limited resolution around the grounding line (Gladstone and others, 2013) was included because it explains the numerical experiment used in this study. We agree that it was confusing to place this section in the introduction, and may give the impression that we advertise using such a friction law. Therefore, we have moved this part to the description of the numerical experiment where the sliding law in question is applied (Sect. 4.2.1), which will also avoid other confusion with respect to the sliding law (technical suggestion p.21).

The manuscript details the coupling between the Full Stokes and Shelfy Stream Approximations, but there are no details about the other difficulties caused by this coupling. Especially, there is no detail on how the domain is discretized into a 2D and a 3D part, and how this division evolves with time (how is the distance to the grounding line computed? how are elements switched between 2D and 3D?). There is also no detail on how the surface evolution is connected into the two parts of the domain. In one part of the domain, only one equation describing the thickness evolution needs to be solved, while in the other part, two equations describing the evolution of the upper and lower surface elevations need to be solved. Similar to the stress balance equation, some explanations must be added to explain the coupling in the surface evolution equations.

Thank you for pointing out missing information, this will be addressed in technical suggestions p.6 l.1-5, p.7 l.21 and p.14 l.16-18.

The notations in the equations are not always consistent, for example between eq.(4) and eq.(6). They both describe a similar quantity but one is based on the components of the tensor, while the other one is based on the velocity derivatives, for no obvious reasons.

Same for $\eta$ and $\bar{\eta}$ in eq.(3) and eq.(5). They both depend on the velocity, but in one case the dependence is explicitly stated while it is not in the other case, which tends to be confusing. There are also many terms introduced that are not necessary, adding more confusion.

Thanks for pointing these inconsistencies out. We have made our notations more consistent, by writing eq. (4) (revised manuscript eq. (5)) in velocity derivatives as well. The velocity dependence is removed from eq. (5) and also from Sect. 3.3 where the viscosity was written with velocity dependence.

Results for the marine ice sheet experiment with a pure Shelfy Stream Approximation should be added to see how different the solution is from a Full Stokes model. The objective here is to assess the algorithm in a case where the Shelfy Stream Approximation and Full Stokes solutions are different. We don't know here if this experiment leads to different results with the two stress balance equations, and therefore there is no evidence that the coupling works in a case where the two solutions are different. So until we see how the coupling works in a case where the Shelfy Stream Approximation and Full Stokes equations lead to significantly different solutions, it is not possible to assess the capability of this coupling method to correctly produce an accurate solution

that needs Full Stokes on a part of the domain.

We show the difference between FS and SSA for the marine ice sheet experiment in comment to the technical suggestion p.12. However, we do not agree that a case where the FS and SSA exhibit a significantly different behavior is necessary to assess the capability of the coupling method. On the contrary, coupling the FS and SSA is only feasible in an area where the FS and SSA are alike, otherwise a coupling cannot provide a continuous velocity field. In cases where part of the domain is such that the FS and SSA exhibit a significantly different behavior, the most important task is to find a suitable coupling location, hence where SSA starts to become applicable.

Technical suggestions:
Note that the line numbering on each page starts with a different number, so I did my best to be clear but it might sometimes induce some confusion.
Thank you, we apologize for the confusion in the numbering.

p.1 l.2: "their non-linearlity" → "the non-linearity"
Done.
p.1 l.2: "are used" → "are commonly used"
Done.
p.1 l.8: "periodical temperature" → "periodic temperature"
Done.
p.1 l.10: "modeling an ice sheet complex" → "modeling a complex ice sheet"
We have changed it to '*a marine ice sheet*'.
p.1 l.15: increased attention to what? (missing words)
Thank you, we have reformulated the first sentences (see next comment).
p.1 l.16: Quantify "much". Also it seems that at least in Greenland, the majority of the changes are caused by surface mass balance changes (Enderlin and others, 2014)
Thank you for pointing this out. As mentioned in reply to the first major comment, we have updated our references and stress that the main *uncertainty* comes from dynamic changes.
p.2 l.1: remove "strongly": some materials rheology are much more non-linear that ice.
Thanks, the equation is strongly non-linear in a mathematical sense, but we agree that it may be confusing to call the rheology strongly non-linear so we have removed "strongly".
p.2 l.7: add Hindmarsh (2004) as a reference for the hybrid models
Thank you again, this reference is added.
p.2 l.13: I disagree with the interpretation of the Pattyn and others (2013) results. In my opinion they show that models including membrane stress and whose grid resolution is sufficiently small capture grounding line evolution in a relatively similar way.
We have rephrased the interpretation of Pattyn and others (2013), to state that it requires inclusion of vertical shearing and not necessarily Full Stokes and also included one more reference (Pattyn and Durand, 2013) to support this interpretation. Also, we have added a reference to MISMIP+.
*"In MISMIP3d, GLD differ between FS models and SSA models, with discrepancies attributed to so-called higher order terms which are neglected in SSA models but included in FS models (Pattyn et al., 2013). Based on these model intercomparisons, it is advised to use models that include vertical shearing to compute reliable projections of ice sheet contribution to sea level rise (Pattyn and Durand, 2013). It should be noted that the experiments in MISMIP3d were idealized, laterally extruded 2D geometries with quite small sideward disturbances and MISMIP+ (Asay-Davis et al., 2016) may give more insight on realistic situations."*

p.2 l.15: "complexes" → "systems"
We have rephrased the sentence, such that this suggestion is not applicable anymore.
p.2 l.17: add a reference for the Ice Sheet System Model
Done, Larour et al. (2012) is added.

p.2 l.23-25: The important question is not so much if friction laws depending on the effective pressure law are faster, but to figure out which ones are more accurate and a allow a good description of the bedrock underlying the ice.

We agreed that the important question is to figure out which sliding law is more accurate. As mentioned in reply to major comment 2, this section was included because it explains the numerical experiment used in this study. To avoid confusion, we have moved this part to the description of the numerical experiment where the sliding law in question is applied (Sect. 4.2.1).

p.3 l.7: "strain rate" → "strain rate tensor"

Done.

p.3 l.15: remove "highly"

Done.

p.3 l.16: I don't see the link between the non-linearity of the Full Stokes model and the derivation of simplified approximations. Most the approximations are also non-linear, and the main purpose of these approximations is to solve a system with less than four coupled unknowns, as is the case with Full Stokes problem.

Agreed, we have reformulated the manuscript,

*"Due to the velocity dependence of the viscosity in Eq. (\ref{eq:Glen}), the FS equations are non-linear. Therefore, many approximations to the FS equations have been derived .. "*

to

*"Due to the velocity dependence of the viscosity in Eq. (\ref{eq:Glen}), the FS equations form a non-linear system with four coupled unknowns, which is time consuming to solve. Therefore, many approximations .."*

p.3 l.20: So what terms are neglected in the Shallow Ice Approximation?

We assume that the Shallow Shelve Approximation is meant here. Thanks for pointing this out, we agree that the manuscript benefits from a more detailed description of the SSA and changed:

*"For ice shelves, the Shallow Shelf Approximation (SSA), has been derived by dimensional analysis based on a small aspect ratio and surface slope (Morland, 1987; MacAyeal, 1989), such that conservation of linear momentum (Eq. (2)) simplifies to "*

to

*"For ice shelves, the Shallow Shelf Approximation (SSA), has been derived by dimensional analysis based on a small aspect ratio and surface slope (Morland, 1987; MacAyeal, 1989). This dimensional analysis shows that vertical variation of u and v is negligible, such that w and p can be eliminated by integrating the remaining stresses over the vertical and applying the boundary conditions at the glacier surface and base. Then, the conservation of linear momentum (Eq. (2)) simplifies to"*

p.4 l.2: How is p eliminated?

See previous comment.

p.4 l.2: I think the main difference that should be explained is that in the Full Stokes case, one has to solve a 3D problem with 4 unknowns, while in the Shallow Shelf Approximation, one has to solve a 2D problem with 2 unknowns.

Agreed, we clarify this by changing

*"The SSA equations are still non-linear through $\bar{\eta}$, but since vertical variation of u and v is neglected, and w and p are eliminated, they are less computationally demanding than FS. "* to

*"The SSA equations are still non-linear through $\bar{\eta}$, but since w and p are eliminated and vertical variation of u and v is neglected, the 3D problem with 4 unknowns is reduced to a 2D problem with 2 unknowns. Therefore, the SSA model is less computationally demanding than FS."*

Fig.1 caption: What is d GL ? Also change "coupling interface" and "grounding line" by "the coupling interface" and "the grounding line"

Thank you for your suggestion, a sentence is added. *"The distance between x_c and x_GL, defined in Eq. (17), is denoted d_GL."*

p.4 l.5: "parametrized" → "represented"
Done.
p.4 Eq.(9): z* does not seem to depend on N in eq.(10).
Thanks for pointing out the confusing notation, the dependence disappears when assuming the hydrostatic balance. We have clarified this by writing *"In line with Gladstone et al. (2017), instead of modeling N, a hydrostatic balance is assumed to approximate z*, .."*
p.5 l.14: "volume gain" is confusing, rephrase.
Rephrased to *"$a_{s/b}$ is the accumulation ($a_{s/b}>0$) or ablation ($a_{s/b}<0$) in meter ice equivalent per year."*
p.5 l.18: Why introduce the ice flux here. This is a new quantity that is never used in the paper, and could be simply replaced by its description (H ū, H v̄)
Agreed, thanks for pointing out this simplification, it is changed in the revised manuscript.
p.5 l.20-23: This 2D/3D explanations are a bit confusing. It is not clear if the SSA part of the model is still in 2D or 3D but the equations are solved only in 1D or 2D on a layer of the model, or if the mesh is indeed changed to 1D or 2D for the SSA, in which case it is not accurate to refer to 2D and 3D models only, and it would be more accurate to say 1D/2D and 2D/3D. Also, there are no details on what is done for the different parts of the mesh.
Thank you for pointing this out, we have rewritten the section such that it is more clear what is meant with 2D and 3D. This section is meant as a theoretical estimate of the memory and performance of a FS-SSA coupling, regardless of the implementation of the coupling, details on what is done for the mesh in the specific coupling presented in the manuscript will be provided in a later section (see comment to p.6 l.1-5 below). We agree that it is not very clear now that we do not consider a specific coupling, and added as a first sentence to this subsection:
*"The reduction of the memory required for a FS-SSA coupling by domain decomposition, compared to a FS model, can be estimated. This estimate is independent of the specific implementation of the coupling between the domains."*

We have rephrased (see revised manuscript):
*"The number of nodes in Ω_FS is then approximately (1 − θ)Nh Nz and in Ω_SSA it is θN_h , neglecting shared nodes on the boundary. For a 3D physical domain, FS and SSA have 4 and 2 unknowns, respectively. Hence, the memory needed to store the solution with a coupled model is proportional to 2Nh (θ + 2(1 − θ)Nz ). For a 2D simulation, where FS has 3 unknowns and SSA only 1, the memory is proportional to Nh (θ + 3(1 − θ)Nz )."*

p.6 l.1-5: As mentioned just above, we don't have any detail on how the discretization of the problem is done in the two parts, and how they are connected. For example, how is the domain decomposed into 2D and 3D (or 1D and 2D), and how does this evolve with time. Details need to be added to understand this part, especially the connection between the two parts of the domain.

Many thanks for emphasizing this, we have added the following:
*"First the velocity $\vec{u}$ (using FS or SSA) is solved for a fixed geometry at time $t$. The mesh always has the same dimension as the physical modeling domain, but $\vec{u}_{SSA}$ is only solved on the basal layer, after which the solution is reprojected over the vertical axis.
Then, the geometry is adjusted by solving the free surface and thickness advection equations using backward Euler time integration."*
Details on the connection between the two parts of the domain are added later, to Section 3.3, see comment to p. 8 l.29.

p.6 l.12: How accurate is the residual free bubbles method?

Different stabilization methods are extensively studied in Gagliardini and Zwinger (2008), and the residual free bubbles method is recommended there, mainly since it provides a better choice on low aspect-ratio elements. This citation is added to the manuscript.

p.7 l.3: "the SSA equations": maybe add here that they are used as Dirichlet boundary conditions right away, that will be more clear.

Thanks for pointing this out, similarly it is also added that the force boundary condition is of Neumann type. We changed *"The FS velocity at $x_c$ provides an inflow boundary condition to the SSA equations." to "The FS velocity at $x_c$ provides a Dirichlet inflow boundary condition to the SSA equations."*

p.7 l.6: What are A and b? Define them.

In other articles, such as Gagliardini et al. (2013), it seems sufficient to define A as the FE system matrix. We have rephrased such that it is more clear that 'system matrix' is a definition in FEM terminology.

*"The SSA equations are linearized, and by means of FEM discretized. This leads to a matrix representation Au = b, where u is the vector of unknown variables (here horizontal SSA velocities). In FEM terminology, the vector b that describes the forces driving or resisting ice flow is usually called the body force and A the system matrix (Gagliardini et al., 2013).¨*

p.7 l.8: What is A SSA ? Define it.

A_SSA is A without the Dirichlet conditions being set, more detailed explained in the next comment.

p.7 l.8: "without the Dirichlet conditions": this is really confusing. It is really hard to follow what is going on here, as none of the terms are explained or detailed.

We agree that information on the way the Dirichlet conditions are set in Elmer/Ice may clarify the derivation of the contact force, therefore we have added

*"In Elmer/Ice, Dirichlet conditions for a node $i$ are prescribed by setting the i'th row of A to zero, except for the diagonal entry which is set to be unity, and $b_i$ is set to have the desired value."*

p.7 l.10: see Appendix A: the derivation of the force applied at the interface of the Full Stokes and Shelfy Stream parts of the domain is the core of the paper. This explanation of where this force comes from and how it was derived need to be entirely moved to the main manuscript, and not hidden in an Appendix.

We prefer to describe the coupling in words in the main text and use more mathematical notation and derivations in the Appendix. Since Referee 2 did not complain about this, we prefer to keep it as it was.

p.7 l.12: Why not take the depth-averaged velocity? This would be more consistent with the Shallow Shelf Approximation that computes a vertically integrated solution.

Yes, we agree that this would be more consistent. However, since the coupling requires a (very close to) constant velocity over the vertical, this choice will not significantly affect the results, but is much more straightforward to implement.

p.7 l.21: The surface evolution is solved differently for the two parts of the domain, but no detail is provided on how these two parts are connected, how the equations are actually solved (together or one after the other), and what is the impact in terms of continuity of the surface elevations, or feedback between the two parts.

Thanks for pointing out this missing information. To ensure continuity, H_SSA (x_c ) = H_FS (x_c )is applied as a boundary condition to the thickness equation. We have added this to the revised manuscript:
*"At x_c , H_SSA = H_FS is applied as a boundary condition to the thickness equation. First the surface evolution is solved for Ω_FS , then Ω_SSA follows."*

p.8 l.29: How are the free surface equations solved to ensure continuity of the solution between the two parts of the domain?
We have added the same information as mentioned in previous reply to the algorithm:
*"Surface evolution by free surface equations (Eq. (14) for Ω_FS*
*Surface evolution by thickness equation (Eq. (15)) for Ω_SSA , with H_SSA (x_c ) = H_FS (x_c )."*

p.10 Fig.3: Consider using round up the numbers in the colorbar. Also "where x c " → "with x c "
Thanks, done.

p.11 l.24: What are the basal conditions applied for this set-up?
We have added a reference to section 2.2.1 where the sliding law is presented, and moved the text on mesh resolution and sliding laws that here, from the introduction:
*"Gagliardini et al. (2016) showed that resolving grounding line dynamics with a FS model requires very high mesh resolution around the grounding line. However, Gladstone et al. (2017) showed that the friction law assumed in this study (see Sect. 2.2.1) reduces mesh sensitivity of the FS model compared to the Weertman friction law assumed in Gagliardini et al. (2016), allowing the coarse mesh used here."*

p.11 l.25: Describe the "SPIN" experiment.
We have added more information about the SPIN experiment (added information in italics).
*"First, the experiment 'SPIN' in Gladstone et al. (2017) is performed, starting from a uniform slab of ice (H=300 m), applying the accumulation in Eq. (22) for 40 kyr, such that a steady state is reached."*

p.11 l.11 "minimal" → "small/limited"
Here, "minimal" did not refer to 30 km (which would make it replaceable by "small"), but to d_GL, since 30 km is not the exact distance before applying SSA, but it is the minimal distance between x_GL and x_c (upon mesh resolution this becomes 32.6 km in this case). We have changed
*"this small difference shows that the minimal distance d GL before applying SSA is sufficient."*
to *"this small difference shows that d _GL=30 km is sufficient."*
p.11 l.11: "For the used mesh resolution" → "For this configuration"
Done.
p.11 l.13: "equal for" → "equal to 30 km for"
Done.
p.11 l.15: Add that the temperature is varied for 500 years and then kept constant for the remaining 2500 years. The equation of the temperature and its description look quite contradictory.
Done.
p.12: results for both diagnostic and prognostic marine ice sheet experiments with a pure SSA model should be added for comparison.
Below, the relative difference [%] between FS and SSA is shown, the relative difference in the grounded part is up to 1.8 percent where u_FS is at least 5 m/yr (not surprising, since basal friction is higher there), on the shelf the relative difference does not exceed 1% (compared to below 0.5% for coupled model). However, we do not consider it necessary to add this result to the manuscript, as argued in response to the last major comment.

[Figure]

p.13 Fig.6 and Fig.7 captions: "solid line" and "dashed line" → "solid lines" and "dashed lines"
Done.
p.14 l.3: "follows" → "comes"
Done.
p.14 l.16-18: For the prognostic experiment, how is the repartition between the two subdomains computed and set-up? Especially, how is the distance to the grounding line for each point computed (in 3D), and how is the mesh changed from 2D to 3D and vice versa when the grounding line evolves. For elements that are switched from the Shallow Shelf Approximation to the Full Stokes approximation, what is used for the velocity at the beginning of the step, especially the vertical velocity?
The mesh is not changed from 2D to 3D, as addressed in comment p.6 l.1-5. We add information to section 3.3 The algorithm, regarding the setup of the domain decomposition:
*"First, the shortest distance d to the grounding line is computed for all nodes in the horizontal footprint mesh at the ice shelf base. Then, a mask is defined that describes whether a node is in $\Omega\_FS$ , $\Omega\_SSA$ or at the coupling interface $x\_c$ , based on the user defined $d\_GL$. Technically, the domain decomposition is solved by the use of passive elements implemented in the overarching Elmer code (Råback et al., 2016), which allow for deactivating and reactivating of elements. An element in $\Omega\_FS$ is declared passive for the SSA solver, such that is not included in the global matrix assembly of $A\_SSA$ , and vice-versa."*
and regarding switching between domains:
*"An element may switch from $\Omega\_SSA$ to $\Omega\_FS$ , for example during grounding line advance. Then, the coupled iteration either starts with the initial condition for $u\_FS$ if the element is in $\Omega\_FS$ for the first time, or the latest $u\_FS (t)$ computed in this element, before it switched to SSA."*
This holds for both horizontal and vertical velocity. The distance is computed based on the coordinates of the nodes in the basal mesh layer, according to Eq. (16) (Eq. (17) in the revised manuscript), by comparing coordinates and minimizing (called DistanceSolver1, the exact code can be found at
https://github.com/ElmerCSC/elmerfem/blob/d2f371327855ba4b73a0038dcc8ecf4400d25e07/fem/src/modules/DistanceSolve.F90 )

p.14 l.5: What is A_FS ? Define it.
Mentioning that it is the system matrix for the FS equations should be enough as definition, as also done in for example Gagliardini et al. (2013).
p.14 l.7: "and and"
Thanks.
p.14 l.19: "the computational work will decrease significantly". This is quite speculative and should be at least replaced by "is expected to"
Replaced.

p.15 l.11: "multiplying with": multiplying what?

The entire equation is multiplied by a test function, this is a standard approach to derive the weak formulation on which the FEM is based. We have rephrased:
*"After multiplying Eq. (2) with a test function v and integrating over the domain \Omega_FS".*

p.16 l.24: "partly coinciding": why partly? There is only one interface between the two subdomains.
Yes, there is only one interface, but strictly speaking they only coinciding at the base, where SSA is solved, not entire on the vertical axis. We add information in parentheses:
*¨partly coinciding with \Gamma_{FSint} (but of one dimension less)"*

p.16 l.11: Eq.(A6) and Eq.(A9) are the same but just rearranged. u_h is not solution of both as it is the same one.
u_h is the finite element approximation of the solution of (A6) and is the solution of (A9) with a particular choice of u_h and v (to be formal).

p.16 l.14: What is f SSA ? Is that how you define it: f SSA = An? In the case why is the first integral over $\Gamma$ SSAint and the second one over $\Gamma$ SSA ? If not, what is f SSA ?
f_{SSA} is defined in Sect 3.1, and now repeated in (A14).
p.17: 20: What about the across flow direction?
Many thanks, we have added information about the lateral boundary condition, denoted as $\Gamma_l$, to the Appendix, below Eq. (A3):
*"Furthermore, there is a lateral boundary $\Gamma_l$ for $\Omega_FS \in R^3$ , where the normal component also vanishes: $v|\Gamma_l \cdot n = 0$ and we assume a vanishing Cauchy-stress vector for unset boundary conditions to velocity components, such that the integral over $\Gamma_l$ vanishes."*
The same holds for the SSA domain, hence the lateral boundary only affects the definition of the test space in Eq. (A7) and the boundary integral vanishes.

p.17 l.22: I don't understand where it comes from. You are trying to estimate the last term of Eq.(A4) to apply this force to the Full Stokes part of the domain. The force applied by one subdomain on the other and vice versa are equal, so instead you try to estimate the second term in eq.(A11). But is it ok to have A instead of σ? And how do you get from the weak form to the local equation This sounds pretty abrupt.
There is a force balance on the boundary at $\Gamma_{FSint}$ and $\Gamma_{SSAint}$. The term on the boundary in (A4) is equal to the term on the boundary in (A11) plus the pressure which is eliminated in SSA. Since SSA is integrated in the z-direction, the force has to be scaled by H.
 If the integrals in the weak form are equal then the forces in the integrals are equal because they are equal for any v, in particular one with local support. A.n corresponds to the part of \sigma.n depending on the velocity u. The pressure part of \sigma.n corresponds to the cryostatic pressure on the SSA boundary.

p.21: There is no mention in the paper of what friction law is applied.
The friction law is found in Eq. (9) (Eq. (10) in the revised manuscript).

**References**
    Asay-Davis, X. S., Cornford, S. L., Durand, G., Galton-Fenzi, B. K., Gladstone, R. M., Gudmundsson, G. H., Hattermann, T., Holland, D. M., Holland, D., Holland, P. R., Martin, D. F., Mathiot, P., Pattyn, F., and Seroussi, H.: Experimental design for three interrelated marine ice sheet and ocean model intercomparison projects: MISMIP v. 3 (MISMIP+), ISOMIP v. 2 (ISOMIP+) and MISOMIP v. 1 (MISOMIP1), Geosci. Model Dev., 9, 2471–2497, doi:10.5194/gmd-9-2471-2016, 2016.
    Gagliardini, O., & Zwinger, T. (2008). The ISMIP-HOM benchmark experiments performed using the Finite-Element code Elmer. *The Cryosphere, 2,* 67-76.

Gagliardini, O., Zwinger, T., Gillet-Chaulet, F., Durand, G., Favier, L., De Fleurian, B., Greve, R., Malinen, M., Martín, C., and Råback, P.: Capabilities and performance of Elmer/Ice, a new-generation ice sheet model, Geosci. Model Dev., 6, 1299–1318, 2013.

Hindmarsh, R.: A numerical comparison of approximations to the Stokes equations used in ice sheet and glacier modeling, J. Geophys. Res.-Earth, 109, 2004.

Larour, E., Seroussi, H., Morlighem, M., and Rignot, E.: Continental scale, high order, high spatial resolution, ice sheet modeling using the Ice Sheet System Model (ISSM), J. Geophys. Res.-Earth, 117, 2012.

Pattyn, F. and Durand, G.: Why marine ice sheet model predictions may diverge in estimating future sea level rise, Geophys. Res. Lett., 40, 4316–4320, 2013.

Pattyn, F., Perichon, L., Durand, G., Favier, L., Gagliardini, O., Hindmarsh, R. C., Zwinger, T., Albrecht, T., Cornford, S., Docquier, D., et al.: Grounding-line migration in plan-view marine ice-sheet models: Results of the ice2sea MISMIP3d intercomparison, J. Glaciol., 59, 410–422, 2013.

Ritz, C., Edwards, T. L., Durand, G., Payne, A. J., Peyaud, V., and Hindmarsh, R. C.: Potential sea-level rise from Antarctic ice-sheet instability constrained by observations, Nature, 528, 115, 2015.

---

## Author Comment (AC4) · 18 Jun 2018

Anonymous Referee #2 comments and response

We would like to thank the reviewer very much for the time and effort put into reviewing our manuscript.  A point-by-point reply (in blue) to each comment (in black) by Referee #2 are given below. Specific changes in the manuscript are written in blue italics.

This work makes modifications to a well-known and widely used glacial flow model Elmer-Ice to allow an approximation to the nonlinear Stokes problem – the Shallow Shelf Approximation – to be solved for a large proportion of the floating part of the domain, in such a way that the two solves are consistent with respect to the force balance. The significance of such a modification is the expense involved with the stokes solve. Not only does it have more degrees of freedom, but it leads to a problem which in the variational sense is a saddle-point and not a minimization problem, leading to difficulty in discretisation; and, though it is not flagged by the authors, there is an extra complication in solving for ice shelves, as a floatation condition cannot be assumed, allowing for oscillation and instability in the vertical momentum condition that must be artificially damped. The work is a nice follow-on slash complement to two other works in the literature: Seroussi et al 2012, which couples together different approximations to the FS problem in a diagnostic setting; and Ahlkrona et al 2016, which dynamically couples FS to SIA in an evolving ice sheet, and is deserving of publication.

We would like to thank you very much for the time and effort you put into reviewing our manuscript.

i only have 2 general comments:

1) My one general comment is that the underlying premise seems to be that this method will reduce computational resource requirements. The reduction in the test cases seems to be low; but this is explained in the discussion, and I am not presently commenting on this. It is rather that presumably this coupling is meant to be applied to regional and continental scale simulations of marine ice sheets. As SSA is only solved on the shelf, the savings are limited by the portion of the domain covered by ice shelves. Quick googling tells me that ice shelves currently represent 10% of Antarctic total area – significantly less than in the test simulations in the paper. So how much more efficient is such a coupling meant to be compared to FS only continental simulations?

When considering the coupling with ISCAL, the gain by having FS-SSA is not just 10% because much of the interior can be modeled with SIA, such that the area left with FS are mainly (around) ice streams and shelves. Also, we are aiming for paleo-simulations, which may have larger ice shelf in the cold period.
We agree that this was not explained in the manuscript yet and have added it to the introduction:
*"The extent of present-day ice shelves is limited to approximately 10 % of the area of Antarctica (Rignot et al., 2013). Therefore, one may question the reduction of computational work by applying SSA to model ice shelves in continental scale simulations of marine ice sheets. However, the coupling is targeted to conducting paleo-simulations, for which much larger ice shelves have been present (Jakobsson et al., 2016; Nilsson et al., 2017). Besides that, the new FS-SSA coupling can be combined with ISCAL. Then, a large part of the interior of a marine ice sheet will be modelled with SIA, such that the FS domain will be restricted to ice streams and areas around the grounding line, when SSA is applied to the ice shelves."*

But I am curious about the issue that i bring up in my first paragraph, but was

not addressed. Durand et al 2009 discusses their approach to prevent instability in the vertical due to ice shelves not being in floatation (they added a degree of implicitness to the velocity solver – their eqn 15). I have wondered if this potential instability affects the solution of FS for a marine ice sheet, and possibly artificially enforces archimedean floatation. I would be curious to know about how your coupling affects this issue ,i.e. the need for the "fix" – could you remove the fix for your coupled experiments?

We have not tried to remove the fix for our coupled experiments. In Elmer/Ice, the Stokes and surface evolution equations are coupled by explicit time stepping, because we only solve the coupled system once per each time step. To solve the fully coupled system implicitly is very costly and unnecessary in most of the cases in ice sheet simulation.
One exception is the flotation, since it is a fast process compared to the dynamics of the upper surface of the ice. As you mention, the term added in Durand et al 2009 indeed behaves as an implicit time stepping for the motion of the lower ice surface in contact with water. In this case, it is not an additional term, but the whole eq (15) is an approximation of the surface equation. One can show that by taking the time step sufficiently small, the error in the position of the lower ice surface is as small as we wish in Elmer/Ice and therefore we have not tried to remove it.

2) There is not much discussion on how the FS and SSA domains are updated dynamically (node reassignment etc). Although you touch on it in the discussion, i would like to see a subsection in the methods section briefly detailing this, even if it makes use of already-existing frameworks. Apologies if such text is there and i overlooked it.

Thanks for pointing this out, indeed the manuscript will improve by adding more information on the technical implementation of the coupling, this information is added to Sect. 3.3 The algorithm:
*"First, the shortest distance d to the grounding line is computed for all nodes in the horizontal footprint mesh at the ice shelf base. Then, a mask is defined that describes whether a node is in $\Omega\_FS$ , $\Omega\_SSA$ or at the coupling interface $x\_c$ , based on the user defined $d\_GL$ . Technically, the domain decomposition is solved by the use of passive elements implemented in the overarching Elmer code (Råback et al., 2016), which allow for deactivating and reactivating of elements.*
*An element in $\Omega\_FS$ is declared passive for the SSA solver, such that is not included in the global matrix assembly of $A\_SSA$ , and vice-versa.*
*An element may switch from $\Omega\_SSA$ to $\Omega\_FS$ , for example during grounding line advance. Then, the coupled iteration either starts with the initial condition for $u\_FS$ if the element is in $\Omega\_FS$ for the first time, or the latest $u\_FS (t)$ computed in this element, before it switched to SSA."*

p2 l 10: MISMIP3D
It is called  MISMIP3d in the title of the paper (Pattyn et al., 2013).

p3 line 7: strain rate tensor
Changed.

p3 l 9: here they are just the Stokes equations. ("Full Stokes" arose because glaciologists realised the equations they had been solving for years were an approximation to Stokes with power-law rheology)
Since our main target public are glaciologists we will keep it like that.

eq 4: your notation for the the 2nd invariant of tensors (in this case D) is a bit subtle, you might consider something else – this is simply a suggestion though
We have written out the 2$^{nd}$ invariant in velocity gradients, Eq. (5) in the revised manuscript.

p3 l 22: "h represents the horizontal components" is a bit ambiguous, and you do not

use the subscript in eq (6)

*We have rewritten 'h represents the components in the x-y plane', and have added the subscript to Eq. (6) (Eq. (7) in the revised manuscript) as well.*

eq (9): is z* a function of effective pressure, which is not defined, or of H as it appears to explicitly be from eq (10)?

*Indeed, it is a function of H, which we clarified by writing*
*"In line with Gladstone et al. (2017), instead of modeling N, a hydrostatic balance is assumed to approximate z*, the dependence on H is written, z*(H)."*

Section 2.3, abbreviations of two dimensions and three dimensions seem awkward, and be clear by 2D you mean the x-z plane.

*Thanks for pointing this out, we have rewritten the section concerning two and three dimensions to:*
*"The number of nodes in $\Omega FS$ is then approximately $(1 - \theta)N_h N_z$ and in $\Omega SSA$ it is $\theta N_h$, neglecting shared nodes on the boundary. For a 3D physical domain, FS and SSA have 4 and 2 unknowns, respectively. Hence, the memory needed to store the solution with a coupled model is proportional to $2N_h (\theta + 2(1 - \theta)N_z)$. For a 2D simulation in the x-z plane, where FS has 3 unknowns and SSA only 1, the memory is proportional to $N_h (\theta + 3(1 - \theta)N_z)$."*

p 5 l 22: say what 3 variables are for FS in 2D.

*After rewriting of the section, this suggestion did not seem applicable anymore.*

p 5 l 23: say what you mean by stokes being a saddle-point problem as many readers will not know what this means.

*We think this it is not necessary to define a saddle-point problem here, but have rephrased:*
*"Furthermore, the FS equations are particularly difficult to solve. In mathematical terminology, they pose a saddle-point problem."*

p 5 l 23: "with all its consequences for numerical treatment" – give examples of these consequences, with references. Most readers will not be familiar with this literature, and others (like me) will be only familiar with some of it.

*We have added "This requires special numerical treatment such as stabilization of the finite element discretization (e.g., Helanow and Ahlkrona, 2018) and special iterative solvers for the resulting system of linear equations (Benzi et al., 2005). By elimination of the pressure, the SSA equations do not form a saddle-point problem."*

p 6 l 25: just \theta, not \theta Nz

*Thanks, changed.*

p6 l 26: say this is approximate, as it must be due to boundaries etc.

*We have added that it is approximate since boundaries are neglected.*

p6 l 6: by "number of unknown variables" do you mean degrees of freedom; or (u,v,w,p) i.e. 4 versus (u,v) i.e. 2? You are saying that, per node, the former takes twice as long to assemble in a matrix versus the latter? I find this difficult to believe in general. What about differing orders of polynomial in the basis functions, and complexity of interaction between DoFs of different variables?

*This is also an approximate expression. If the order of the methods is the same and the complexity of the interaction between the components is similar then the formula is an estimate. Changed to:*
*"The work to assemble the matrices grows linearly with the number of unknown variables. Suppose that this work for FS in 3D is $4C_{FS}N_hN_z$ in the whole domain, for FS $4C_{FS}(1-\theta)N_hN_z$ in $\Omega_{FS}$, and for SSA $2C_{SSA}\theta N_h$ in $\Omega_{SSA}$. The coefficients $C_{FS}$ and $C_{SSA}$ depend on the basis functions for FS and SSA and the*

*complexity of the equations. The reduction in assembly time for the matrix is $q_{ass}=1-\theta+C_{SSA}\theta/2C_{FS}N_z$. If $C_{FS}\approx C_{SSA}$ then the reduction is approximately as in (16). The same conclusion holds in 2D. Therefore, the reduction of that part is estimated to be similar to the reduction in Eq (15)."*

p 6 l 16: is there a theoretical basis for d_{GL} – such as the estimate of the non-hydrostatic boundary layer from Schoof (2011, JFM)? Or arbitrary?

We have no theoretical basis for d_{GL}. As mentioned in the discussion we propose that further studies let Ω SSA be determined automatically, not necessarily based on d_{GL} but for example based on a tolerance for the vertical variation of the horizontal velocities (that should be close to zero in order to allow for a smooth coupling to SSA) or by using a posteriori error estimates based on the residual as derived in Jouvet (2016).

p 7 eq (17): this is an initial guess for the membrane stress at the interface, yes? Would a better one not be \sigma_{FS} \cdot \vec{n} = \rho_w g (-z) if z<0, i.e. the condition which would be applied if xc was a calving front? (This would be exact in the 2D case with no buttressing seaward of x_c; and i wonder if it would reduce iteration count in general)

Good point, probably it would reduce iterations, we have considered this initial guess during the implementation as well. However, this would only be beneficial during the first time step, or when the coupling interface x_c has changed position. Any other time, the algorithm will take f_SSA from the previous iteration and this will thus be a good initial guess.

p 9 l 12: "assembly time ... almost doubles" – i think this is the issue you address in the discussion? would be good to say that it is addressed in the discussion, as i was confused by this when i read it.

Thanks for pointing this out, we have added *"This issue is due to usage of passive elements and is addressed in the Discussion (Sect. 5)."*

p 9 l 15: is v velocity in the y-direction? better say so. aside from no normal flow, what is the other BC at the lateral boundary? no stress? no flow?

We have changed $v=0$ to $u\cdot n=0$. The unset boundary conditions for remaining velocity components, by the natural boundary condition resulting from a partial integration of the stress divergence in the weak formulation, automatically apply a vanishing Cauchy-stress vector in that direction. This information has been added to the Appendix,  below Eq. (A3):
*"Furthermore, there is a lateral boundary $\Gamma_l$ for $\Omega_{FS} \in R^3$ , where the normal component also vanishes: $v|\Gamma_l \cdot n = 0$ and we assume a vanishing Cauchy-stress vector for unset boundary conditions to velocity components, such that the integral over $\Gamma_l$ vanishes."*

p 11 line 8: 58% of the nodes – you mean in the projection of the grid to the x-y plane?

Yes, thanks for pointing this out, changed to "58% of the nodes in the horizontal footprint mesh are located inside \Omega_{SSA} (θ = 0.58)".

p 12 paragraph at line 13: found this discussion a bit difficult to follow – is there any way the main performance points can be summarised in a table? also limited –> varied

Thanks for pointing this out, we have added a table and reduced the amount of information in this paragraph, referring to the table instead.

p 14 discussion at line 10: if you were to make this change re: node assignment at a lower level, how would it affect the ability to easily update the Full Stokes sub-domain?

This would not affect the ability to easily update the Full Stokes sub-domain, it would be based on the same mask as the passive/active updates are done now (see the extra information added in Sect. 3.3, in reply to the second general comment). It was shown for ISCAL that this can be done efficient and dynamically in Elmer/Ice (Ahlkrona et al. 2016).

p 16 line 24: you previously used A as a symbol for a matrix

Thanks, changed to B such that there is no confusion with A being previously defined as a system matrix. However, B (or A in old version)  is a matrix, with derivatives in the elements.

p 16 line 4: did you define f_{CF}?

We had only given the FS version of the boundary condition at the calving front ( Eq. (12), (13) in the revised manuscript) and added: *"f_{CF}, as in Eq. (13) but integrated over z"* to the Appendix.

p 16 eq (A7) and below: suggest to use w as a test function symbol

We chose to stick to v for a test function.

p 16 eq (A9) the 1st boundary integral is over \Gamma_{SSA} not \Gamma_{SSAint}

Indeed, updated.

**References**

Ahlkrona, J., Lötstedt, P., Kirchner, N., & Zwinger, T. (2016). Dynamically coupling the non-linear Stokes equations with the Shallow Ice Approximation in glaciology: Description and first applications of the ISCAL method. *Journal of Computational Physics*, *308*, 1-19.

Benzi, M., Golub, G. H., and Liesen, J.: Numerical solution of saddle point problems, Acta Numer., 14, 1–137, 2005.

Helanow, C. and Ahlkrona, J.: Stabilized equal low-order finite elements in ice sheet modeling–accuracy and robustness, Computat. Geosci., pp. 1–24, 2018.

Jakobsson, M., Nilsson, J., Anderson, L., Backman, J., Björk, G., Cronin, T. M., Kirchner, N., Koshurnikov, A., Mayer, L., Noormets, R., et al.: Evidence for an ice shelf covering the central Arctic Ocean during the penultimate glaciation, Nat. Commun., 7, 10 365, 2016.

Jouvet, G.: Mechanical error estimators for shallow ice flow models, J. Fluid Mech., 807, 40–61, 2016

Nilsson, J., Jakobsson, M., Borstad, C., Kirchner, N., Björk, G., Pierrehumbert, R. T., and Stranne, C.: Ice-shelf damming in the glacial Arctic Ocean: dynamical regimes of a basin-covering kilometre-thick ice shelf, The Cryosphere, 11, 1745, 2017.

Pattyn, F., Perichon, L., Durand, G., Favier, L., Gagliardini, O., Hindmarsh, R. C., Zwinger, T., Albrecht, T., Cornford, S., Docquier, D., et al.: Grounding-line migration in plan-view marine ice-sheet models: Results of the ice2sea MISMIP3d intercomparison, J. Glaciol., 59, 410–422, 2013.

---

## Referee Report (RR1)

Review of "Dynamically coupling Full Stokes and Shallow Shelf Approximation for marine ice sheet flow using Elmer/Ice (v8.3)"

**1 Summary statement**

This manuscript is much improved compared to the previous version. I especially like that the authors clarified several points concerning the resolution of the mass balance equation or the treatment of Stokes and Shallow Shelf Approximation elements in the mesh. There are a few points (listed below) that I think should be nuanced or better explained. The last point on the derivation of the force in the Appendix is especially important I think for people being able to understand and reproduce this work.

**2 Specific comments**

The page and line numbers refer to the manuscript with including the differences for this new version.

p.2 l.11-20: I think this paragraph should be nuanced: we don't know how much of the difference is caused by the difference in the stress balance approximation used, and how much is due to the different treatment of the grounding line problem (contact versus hydrostatic equilibrium).

p.6 l.19: I don't really understand why the reduction of memory is independent of the coupling implementation. For example, here "ghost" nodes are created for Stokes even when the Shallow Shelf Approximation is used. Different choices would lead to different memory requirements, so it seems that the choices made for the coupling impact the memory requirements.

p.8 l.18: remove (Gagliardini et al., 2013) as this is generic to the finite element method and not specific to Elmer/Ice.

p.9 l.22-29: So what are the criteria used to stop the iterations for the Stokes iterations, the Shallow Shelf iterations, and the coupled iterations?

Response to reviewer: the numbers provided for the difference between the Stokes and Shallow Shelf Approximation solutions for the prognostic case should be added to the text. I agree that the figure does not add much, but these numbers are important. Also, a 1.8% maximum difference between the two solutions is really small. The difference between Stokes and the coupled solution is much reduced, but there must be a simple

test that provides larger differences between Stokes and Shallow Shelf Approximation. The introduction emphasizes the importance of using Stokes, so an example showing that would be appropriate.

p.19 Eq.A14: Thanks for clarifying the Appendix, it is now easier to follow. However, I am not sure to understand the last step leading to Eq.A14. From Eq.A13 and using the information in lines 19-23, I still don't understand how you go from the integrated form to Eq.A14. You are left with the term in A11 equal to the first term of Eq.12 integrated over $\Gamma_{SSAint}$. How do you go back to a regular equation given that one term has v and the other one $\nabla_h v$? I probably missed something, so it would be great to add an intermediate step before Eq.A14.

---

## Author Response (AR2)

Review of "Dynamically coupling Full Stokes and Shallow Shelf Approximation for marine ice sheet flow using Elmer/Ice (v8.3)"

We would like to thank the reviewer for the time and effort put into reviewing our revised manuscript. A point-by-point reply (in blue) to each comment (in black) by Referee #1 are given below. Specific changes in the manuscript are written in red and a marked-up manuscript version follows after this reply.

1 Summary statement
This manuscript is much improved compared to the previous version. I especially like that the authors clarified several points concerning the resolution of the mass balance equation or the treatment of Stokes and Shallow Shelf Approximation elements in the mesh. There are a few points (listed below) that I think should be nuanced or better explained. The last point on the derivation of the force in the Appendix is especially important I think for people being able to understand and reproduce this work.

Thank you, we have replied to the specific points below and added more explanation of the derivation to the Appendix.

2 Specific comments
The page and line numbers refer to the manuscript with including the differences for this new version.

p.2 l.11-20: I think this paragraph should be nuanced: we don't know how much of the difference is caused by the difference in the stress balance approximation used, and how much is due to the different treatment of the grounding line problem (contact versus hydrostatic equilibrium).
We have added this nuance to the paragraph:

"Despite many recent efforts, modelling GLD still poses a challenge in numerical models, as illustrated by the wide range of results obtained in the Marine Ice Sheet Model Intercomparison Project (MISMIP, Pattyn et al., 2012) . In MISMIP3d, GLD differ between FS models and SSA models, with discrepancies attributed to so-called higher order terms which are neglected in SSA models but included in FS models (Pattyn et al., 2013). Based on these model intercomparisons, it is advised to use models that include vertical shearing to compute reliable projections of ice sheet contribution to sea level rise (Pattyn and Durand, 2013) . On the other hand, it is not entirely clear how much of the difference in GLD is due to the different numerical treatment of the grounding line problem in shallow models. An updated version of the hybrid SIA/SSA Parallel Ice Sheet Model (PISM) that uses a modified driving stress calculation and subgrid grounding line interpolation showed GLD comparable to a FS model (Feldmann et al., 2014). It should be noted that the experiments in MISMIP3d were idealized, laterally extruded 2D geometries with quite small sideward disturbances and MISMIP+ (Asay-Davis et al., 2016) may give more insight on realistic situations. Additionally, there is a recent publication that sheds new light to a possible problem with the setup of MISMIP experiments (Gladstone et al., 2018). "

p.6 l.19: I don't really understand why the reduction of memory is independent of the coupling implementation. For example, here "ghost" nodes are created for Stokes even when the Shallow Shelf Approximation is used. Different choices would lead to different memory requirements, so it seems that the choices made for the coupling impact the memory requirements.

Thank you for pointing out the need of clarification here. We only consider the reduction of memory in the most ideal case of an implementation without ghost nodes, which can be seen as an estimate of the upper bound for reduction of memory. This is added to the manuscript:

"The reduction of the memory required for a FS-SSA coupling by domain decomposition, compared to a FS model, can be estimated. This estimate is independent of the specific implementation of the coupling between the domains and concerns only the most ideal implementation in which no redundant information is stored."
There are additional terms coming from the coupling surface (or line in 2D) (with or without ghost cells) but these terms are much smaller than those from the volume (area in 2D) and are therefore neglected to simplify the expressions, as mentioned in the manuscript ("neglecting shared nodes on the boundary").

p.8 l.18: remove (Gagliardini et al., 2013) as this is generic to the finite element method and not specific to Elmer/Ice.
Ok.

p.9 l.22-29: So what are the criteria used to stop the iterations for the Stokes iterations, the Shallow Shelf iterations, and the coupled iterations?
They are given in Tables A1, A2.

Response to reviewer: the numbers provided for the difference between the Stokes and Shallow Shelf Approximation solutions for the prognostic case should be added to the text. I agree that the figure does not add much, but these numbers are important. Also, a 1.8% maximum difference between the two solutions is really small. The difference between Stokes and the coupled solution is much reduced, but there must be a simple test that provides larger differences between Stokes and Shallow Shelf Approximation. The introduction emphasizes the importance of using Stokes, so an example showing that would be appropriate.
Thank you, we agree that the numbers should be added. We added this to Section 4.2.1:

"This removal of buttressing leads to grounding line retreat from 871.2 km to 730.8 km (Fig. 4). Again, FS-only, SSA-only and the coupled model are applied to this setup. Where $u_{FS} \geq 5$ m yr$^{-1}$, the relative difference between $u_{FS}$ and $u_{SSA}$ is below 1.8%. The velocity $u_c$ is given in Fig. 4, with $d_{GL} = 30$ km such that 58% of the nodes in the horizontal footprint mesh are located inside $\Omega_{SSA}$ ($\theta = 0.58$)."
However, we do not agree that there 'there must be a simple test that provides larger differences between Stokes and Shallow Shelf Approximation'. The goal of study is not to show importance of using Stokes. Indeed, we do emphasize the importance of using Stokes as motivation for the current study, but we think that previously published literature is sufficiently convincing.

p.19 Eq.A14: Thanks for clarifying the Appendix, it is now easier to follow. However, I am not sure to understand the last step leading to Eq.A14. From Eq.A13 and using the information in lines 19-23, I still don't understand how you go from the integrated form to Eq.A14. You are left with the term in A11 equal to the first term of Eq.12 integrated over $\Gamma$ SSAint . How do you go back to a regular equation given that one term has v and the other one $\nabla_h v$? I probably missed something, so it would be great to add an intermediate step before Eq.A14.

We agree that it is a big step to get from Eq. A13 to Eq. A14, however, a more detailed explanation would include the entire derivation of the SSA equations. It can be compared to the derivation of $f_{CF}$ in Eq. A12, the calving front boundary condition for SSA.

For a FS model, this is simply the hydrostatic pressure as in Eq. (13):

$$\boldsymbol{\sigma} \cdot \boldsymbol{n} = -p_w \boldsymbol{n} \text{ where } p_w(z) = -\rho_w g z \text{ if } z \leq 0$$

However, for SSA, when using FEM, the force at the boundary is prescribed as

$$\mathbf{f_{CF}} = \int_{z_b}^{z_s} \rho g(z_s - z) - p_w(z) \mathrm{d}z = \frac{\rho g}{2} H^2 - \frac{\rho_w g}{2} z_b^2$$

assuming that the sea level is at 0. The above equation is often simplified using the floatation criterion (see eg Greve and Blatter, 2009) such that

$$\mathbf{f_{CF}} = \frac{\rho}{2\rho_w}(\rho_w - \rho)g H^2.$$

We have added an equation to the Appendix (A14) that shows how the stress applied to the FS equations is written in terms of $f_{SSA}$, compensating for the cryostatic pressure and vertical integration inversely.

**References**

[revised manuscript text omitted]